# Clinical implications of bone marrow adiposity identified by phenome-wide association and Mendelian randomization in the UK Biobank

Wei Xu [1,2], Ines Mesa-Eguiagaray[1], David M. Morris[2,3], Chengjia Wang [3,4], Calum D. Gray [3], Samuel Sjöström[2], Giorgos Papanastasiou[3,5], Sammy Badr [6], Julien Paccou[6], Lijuan Wang [1], Xue Li[7], Paul R. H. J. Timmers [8], Maria Timofeeva [8,9], Scott IK Semple[2,3], Tom MacGillivray [10], Evropi Theodoratou [1,11] ✉ & William P. Cawthorn [2] ✉

Bone marrow adiposity changes in diverse diseases, but the full scope of these, and whether they are directly influenced by marrow adiposity, remains unknown. To address this, we previously measured the bone marrow fat fraction of the femoral head, total hip, femoral diaphysis, and spine of over 48,000 UK Biobank participants. Here, we first use these data for PheWAS to identify diseases associated with marrow adiposity at each site. This reveals associations with 47 incident diseases across 12 disease categories, including osteoporosis, fracture, type 2 diabetes, cardiovascular diseases, cancers, and other conditions that burden public health worldwide. Intriguingly, type 2 diabetes associates positively with spine bone marrow adiposity but negatively with marrow adiposity at femoral sites. We then establish PRSs based on bone-marrow-fat-fraction-associated SNPs and use PRS-PheWAS and Mendelian randomization to explore causal associations between marrow adiposity and disease. PRS-PheWAS reveals that genetic predisposition to increased marrow adiposity is positively associated with osteoporosis and fractures. Mendelian randomization further suggests that increased marrow adiposity at the diaphysis and total hip is causally associated with osteoporosis. Our findings substantially advance understanding of how marrow adiposity impacts human health and highlight its potential as a biomarker and/or therapeutic target for diverse human diseases.

Bone marrow is a significant fat storage site across species, ranging from fish to mammals[1]. Adipocytes are a normal component of the bone marrow microenvironment and collectively form bone marrow adipose tissue (BMAT), which comprises over 70% of bone marrow volume and ~10% of total fat mass in lean, healthy adults[2]. BMAT further increases with age and in various pathological and iatrogenic conditions, including osteoporosis, obesity, type 2 diabetes (T2D), radiotherapy, and glucocorticoid exposure[2–4]. Unlike other adipose depots, BMAT also expands in energy-deficient states such as anorexia nervosa and during controlled caloric restriction[2,5–8]. The molecular

**Fig. 1 | Study design. A** Deep learning was used to segment the spine, femoral head, total hip, and femoral diaphysis from MRI scans of participants in the UKBB imaging study, allowing BMFF measurements at each site. Genetic associates of BMFF were identified by genome-wide association meta-analyses (GWAS), from which polygenic risk scores (PRS) were established for BMFF at each site. **B, C** PheCODES were used to classify disease outcomes for phenome-wide association studies (PheWAS). The observational PheWAS (Obs-PheWAS) (**B**) used BMFF as the predictor, while PRS-PheWAS (**C**) used PRS as the predictor. **D** Mendelian Randomization was done, including the Bayesian MR-HORSE method, to assess causal relationships between BMFF at each site and osteoporosis, fracture, and type 2 diabetes. Figure generated using Adobe Illustrator. Images in (**A**) are reused from our previous publications[13,15] under a CC-BY license. Some graphics in (**D**) are adapted from www.svgrepo.com under an open license.

**Table 1 | Summary of the characteristics of the Obs-PheWAS samples for the four bone regions**

| | Femoral head (n = 45,288) | Total hip (n = 45,187) | Diaphysis (n = 44,099) | Spine (n = 48,427) |
|---|---|---|---|---|
| Age at imaging (years)* | 65.13 (7.79) | 65.26 (7.79) | 65.07 (7.77) | 65.13 (7.79) |
| *Sex* | | | | |
| Male | 21,100 (46.59%) | 20,510 (45.39%) | 19,813 (44.93%) | 23,461 (48.45%) |
| Female | 24,188 (53.41%) | 24,677 (54.61%) | 24,286 (55.07%) | 24,966 (51.55%) |
| *Ancestry* | | | | |
| White | 45,288 (100.00%) | 45,187 (100.00%) | 44,099 (100.00%) | 48,427 (100.00%) |
| BMI at imaging (kg/m$^2$)* | 26.62 (4.31) | 26.43 (4.33) | 26.59 (4.34) | 26.57 (4.36) |
| *BMFF measurement* | | | | |
| BMFF (%) | 91.16 (2.82) | 90.49 (3.60) | 82.09 (4.83) | 51.21 (8.22) |
| Size of segmented region (voxels) | 750.70 (277.97) | 1051.00 (417.03) | 92.78 (22.92) | 2106.00 (466.69) |
| Rank-transformed BMFF | 3.01E-13 (0.99) | 2.55E-14 (0.99) | 7.28E-13 (0.99) | 1.15E-13 (0.99) |

*Values are presented as mean (SD). Numbers of participants in each sex and ancestry category are shown as absolute numbers, with % of total for each bone region indicated in parentheses. *BMI* body mass index, *BMFF* bone marrow fat fraction.

and functional characteristics of BMAT vary based on its skeletal location, with distinct differences between axial and appendicular bones[2]. Together, these observations show that BMAT is a normal feature of vertebrate anatomy that is altered in diverse pathophysiological contexts and has site-specific properties. However, compared to the study of white and brown adipose tissues, research on BMAT has been surprisingly limited[2]. Thus, the physiological and pathological roles of BMAT remain largely unknown.

Several methods allow non-invasive measurement of bone marrow adiposity. The gold standards are magnetic resonance imaging (MRI) with chemical shift-encoding for water–fat separation, and magnetic resonance spectroscopy, which can each quantify the bone marrow fat fraction (BMFF)[9,10]. This has been done in numerous small-scale human cohort studies, providing some insights into how BMAT changes in specific pathophysiological contexts. A particular focus has been on ageing and related conditions, especially skeletal and metabolic diseases[3,11]. For instance, higher bone marrow adiposity at the lumbar spine has been linked to morphometric vertebral fractures and lower bone mineral density (BMD) in both osteoporotic and non-osteoporotic individuals[3,11,12]. However, these observational studies have never analyzed more than 729 participants[13] and typically follow a case–control design for specific diseases, limiting the ability to detect other associations. Critically, whether BMAT directly influences the development and progression of musculoskeletal, metabolic or other diseases remains unknown.

To address these limitations, we developed a deep-learning algorithm for large-scale analysis of BMFF in the UK Biobank (UKBB). In what is the world's largest health imaging study, 100,000 UKBB participants are undergoing MRI of the brain, heart and whole body, as well as dual-energy X-ray absorptiometry (DXA) to measure BMD[14]; imaging of all 100,000 participants was completed in July 2025. Our deep-learning method allows accurate and efficient measurement of BMFF within the spine, femoral head, total hip, and femoral diaphysis[13,15]. These four sites cover the axial and appendicular skeleton and include major sites of fracture burden, ensuring that the BMFF measurements allow detection of site-specific and clinically relevant BMAT characteristics.

We have since applied our deep-learning models to measure BMFF of the spine, femoral head, total hip, and femoral diaphysis in over 44,000 individuals[15] and used these data for genome-wide association analyses (GWAS) to identify the genetic variants associated with altered BMFF at each site[15]. Moreover, we established the associations between BMFF and age, body mass index (BMI), BMD, regional adiposity, and other physiological and anatomical traits[15]. Intriguingly, these genetic and anatomical associations often differ for BMFF across the four skeletal sites, underscoring the site-specific nature of BMAT formation and function.

Herein, we use our site-specific BMFF measurements and polygenic risk scores (PRSs) derived from our GWAS to perform phenome-wide association studies (PheWAS), thereby comprehensively identifying diseases associated with altered BMFF. Finally, we use Mendelian randomization analysis (MR) to establish evidence of causality between altered BMFF and human diseases. Together, these findings represent a groundbreaking advance in our understanding of the clinical relevance of BMAT.

## Results
The study design is presented in Fig. 1.

### Phenome-wide association analyses
A total of 10,750 unique International Classification of Diseases-10 (ICD-10) and 3113 ICD-9 codes were summarized from hospital inpatient, cancer registry, and death registry data of the UKBB cohort. The ICD codes were mapped into PheCODE groups[16], which included 1808 PheCODEs classified into 17 disease categories [e.g., genitourinary diseases = 173 (9.57%); circulatory diseases = 171 (9.46%); endocrine/metabolic diseases = 169 (9.35%); digestive diseases = 162 (8.96%); neoplasms = 140 (7.74%); musculoskeletal diseases = 132 (7.30%); injuries & poisonings = 120 (6.64%); dermatologic diseases = 95 (5.25%); neurological diseases = 84 (4.65%)].

### Observational –PheWAS (Obs-PheWAS)
In Obs-PheWAS, the exposure was the rank-transformed (normalized) BMFF for each bone region, as measured by our deep-learning models[13,15]. The outcomes were PheCODES for incident disease cases diagnosed at least six months after participants underwent the MRI scan. Obs-PheWAS sample quality control is shown in Supplementary Data 1, while the PheCODEs used for Obs-PheWAS are summarized in Supplementary Data 2. Associations were considered statistically significant after applying FDR correction. The baseline characteristics of the Obs-PheWAS sample for each bone region are summarized in Table 1 (femoral head: N = 45,288; total hip: N = 45,187; diaphysis: N = 44,099; spine: N = 48,427).

In the femoral head Obs-PheWAS, BMFF was associated with increased risk of osteoporosis and 'other disorders of bone and cartilage', but with decreased risk of acute renal failure and osteoarthritis (Table 2; Fig. 2).

In the total hip Obs-PheWAS, eleven PheCODEs, spanning nine disease groups, were associated with BMFF (Table 2; Fig. 2). Among

**Table 2 | Incident disease outcomes associated with BMFF in the Obs-PheWAS**

| Phe CODE | Description | Group | Cases (n) | Controls (n) | Beta ± SE | OR (95% CIs) | P value | Q value | Bonferroni |
|---|---|---|---|---|---|---|---|---|---|
| **Femoral head (n = 4)** | | | | | | | | | |
| 743.1 | Osteoporosis NOS | musculoskeletal | 472 | 44,090 | 0.42 ± 0.06 | 1.52 (1.36–1.71) | 7.73E-13 | 8.81E-11 | TRUE |
| 585.1 | Acute renal failure | genitourinary | 506 | 43,590 | -0.19 ± 0.05 | 0.83 (0.75–0.91) | 8.79E-05 | 5.01E-03 | TRUE |
| 733 | Other disorders of bone and cartilage | musculoskeletal | 259 | 43,035 | 0.27 ± 0.08 | 1.31 (1.13–1.52) | 3.77E-04 | 1.43E-02 | TRUE |
| 740.1 | Osteoarthritis; localized | musculoskeletal | 567 | 42,116 | -0.15 ± 0.05 | 0.86 (0.78–0.94) | 1.31E-03 | 3.73E-02 | FALSE |
| **Total hip (n = 11)** | | | | | | | | | |
| 743.1 | Osteoporosis NOS | musculoskeletal | 495 | 43,933 | 0.53 ± 0.05 | 1.69 (1.52–1.88) | 8.19E-23 | 9.33E-21 | TRUE |
| 994.2 | Sepsis | injuries & poisonings | 266 | 44,742 | -0.34 ± 0.07 | 0.71 (0.62–0.81) | 5.57E-07 | 3.18E-05 | TRUE |
| 38 | Septicemia | infectious diseases | 270 | 43,572 | -0.32 ± 0.07 | 0.72 (0.63–0.83) | 1.58E-06 | 6.00E-05 | TRUE |
| 250.2 | Type 2 diabetes | endocrine/metabolic | 598 | 42,864 | -0.21 ± 0.05 | 0.81 (0.74–0.89) | 4.41E-06 | 1.26E-04 | TRUE |
| 174.1 | Malignant neoplasm of the female breast | neoplasms | 301 | 41,962 | -0.3 ± 0.07 | 0.74 (0.64–0.85) | 1.25E-05 | 2.84E-04 | TRUE |
| 585.1 | Acute renal failure | genitourinary | 508 | 43,506 | -0.2 ± 0.05 | 0.82 (0.75–0.91) | 6.63E-05 | 1.26E-03 | TRUE |
| 733 | Other disorders of bone and cartilage | musculoskeletal | 268 | 42,885 | 0.28 ± 0.07 | 1.32 (1.15–1.52) | 8.51E-05 | 1.39E-03 | TRUE |
| 401.1 | Essential hypertension | circulatory system | 2845 | 35,114 | -0.08 ± 0.02 | 0.92 (0.88–0.96) | 1.77E-04 | 2.53E-03 | TRUE |
| 278.1 | Obesity | endocrine/metabolic | 938 | 43,154 | -0.14 ± 0.04 | 0.87 (0.81–0.94) | 2.63E-04 | 3.34E-03 | TRUE |
| 496 | Chronic airway obstruction | respiratory | 297 | 40,905 | -0.19 ± 0.06 | 0.83 (0.73–0.94) | 3.70E-03 | 4.22E-02 | FALSE |
| 574.1 | Cholelithiasis | digestive | 215 | 42,954 | -0.22 ± 0.08 | 0.81 (0.69–0.94) | 4.80E-03 | 4.98E-02 | FALSE |
| **Diaphysis (n = 2)** | | | | | | | | | |
| 743.1 | Osteoporosis NOS | musculoskeletal | 473 | 42,916 | 0.49 ± 0.05 | 1.63 (1.47–1.81) | 1.04E-19 | 1.19E-17 | TRUE |
| 250.2 | Type 2 diabetes | endocrine/metabolic | 587 | 41,818 | -0.15 ± 0.04 | 0.87 (0.8–0.94) | 8.06E-04 | 4.60E-02 | FALSE |
| **Spine (n = 40)** | | | | | | | | | |
| 496 | Chronic airway obstruction | respiratory | 318 | 43,836 | 0.47 ± 0.06 | 1.59 (1.41–1.79) | 1.94E-14 | 2.45E-12 | TRUE |
| 250.2 | Type 2 diabetes | endocrine/metabolic | 656 | 45,866 | 0.33 ± 0.04 | 1.39 (1.27–1.51) | 4.80E-14 | 3.02E-12 | TRUE |
| 272.1 | Hypercholesterolemia | endocrine/metabolic | 1678 | 42,738 | 0.2 ± 0.03 | 1.22 (1.16–1.29) | 2.24E-13 | 9.39E-12 | TRUE |
| 743.1 | Osteoporosis NOS | musculoskeletal | 502 | 47,145 | 0.35 ± 0.05 | 1.41 (1.28–1.56) | 1.55E-11 | 4.89E-10 | TRUE |
| 306 | Other mental disorder | mental disorders | 2108 | 41,114 | 0.16 ± 0.03 | 1.18 (1.12–1.23) | 5.20E-11 | 1.31E-09 | TRUE |
| 272.1 | Hyperlipidemia | endocrine/metabolic | 410 | 42,738 | 0.32 ± 0.05 | 1.37 (1.23–1.53) | 4.64E-09 | 9.74E-08 | TRUE |
| 530.1 | GERD | digestive | 1328 | 41,524 | 0.17 ± 0.03 | 1.19 (1.12–1.26) | 2.68E-08 | 4.82E-07 | TRUE |
| 411.8 | Other chronic ischemic heart disease, unspecified | circulatory system | 616 | 44,325 | 0.22 ± 0.04 | 1.25 (1.15–1.36) | 4.45E-07 | 7.01E-06 | TRUE |
| 721.1 | Spondylosis without myelopathy | musculoskeletal | 405 | 46,047 | 0.27 ± 0.06 | 1.31 (1.18–1.46) | 6.84E-07 | 9.58E-06 | TRUE |
| 366 | Cataract | sense organs | 1522 | 43,169 | 0.14 ± 0.03 | 1.15 (1.08–1.22) | 2.38E-06 | 3.00E-05 | TRUE |
| 211 | Benign neoplasm of other parts of digestive system | neoplasms | 413 | 46,420 | 0.25 ± 0.05 | 1.28 (1.15–1.42) | 4.94E-06 | 5.66E-05 | TRUE |
| 318 | Tobacco use disorder | mental disorders | 275 | 44,852 | 0.29 ± 0.07 | 1.33 (1.17–1.52) | 1.01E-05 | 1.06E-04 | TRUE |
| 562.1 | Diverticulosis | digestive | 1884 | 36,758 | 0.11 ± 0.03 | 1.12 (1.06–1.18) | 1.33E-05 | 1.29E-04 | TRUE |
| 411.2 | Myocardial infarction | circulatory system | 499 | 44,325 | 0.21 ± 0.05 | 1.23 (1.12–1.35) | 2.33E-05 | 2.10E-04 | TRUE |
| 366.2 | Senile cataract | sense organs | 1442 | 43,169 | 0.13 ± 0.03 | 1.14 (1.07–1.2) | 2.52E-05 | 2.12E-04 | TRUE |
| 785 | Abdominal pain | symptoms | 851 | 42,666 | 0.15 ± 0.04 | 1.16 (1.08–1.25) | 6.76E-05 | 5.03E-04 | TRUE |
| 411.3 | Angina pectoris | circulatory system | 514 | 44,325 | 0.19 ± 0.05 | 1.21 (1.1–1.33) | 6.79E-05 | 5.03E-04 | TRUE |
| 208 | Benign neoplasm of colon | neoplasms | 1410 | 43,838 | 0.12 ± 0.03 | 1.12 (1.06–1.19) | 1.03E-04 | 7.23E-04 | TRUE |
| 720 | Spinal stenosis | musculoskeletal | 262 | 46,047 | 0.26 ± 0.07 | 1.29 (1.13–1.47) | 1.86E-04 | 1.23E-03 | TRUE |

**Table 2 (continued) | Incident disease outcomes associated with BMFF in the Obs-PheWAS**

| Phe CODE | Description | Group | Cases (n) | Controls (n) | Beta ± SE | OR (95% CIs) | P value | Q value | Bonferroni |
|---|---|---|---|---|---|---|---|---|---|
| 300.1 | Anxiety disorder | mental disorders | 707 | 41,114 | 0.16 ± 0.04 | 1.17 (1.08–1.27) | 2.21E-04 | 1.39E-03 | TRUE |
| 585.3 | Chronic renal failure [CKD] | genitourinary | 236 | 46,610 | 0.26 ± 0.07 | 1.3 (1.13–1.49) | 2.42E-04 | 1.45E-03 | TRUE |
| 418 | Nonspecific chest pain | circulatory system | 641 | 44,259 | 0.15 ± 0.04 | 1.16 (1.07–1.26) | 6.10E-04 | 3.50E-03 | FALSE |
| 733 | Other disorders of bone and cartilage | musculoskeletal | 269 | 46,031 | 0.23 ± 0.07 | 1.26 (1.1–1.44) | 7.80E-04 | 4.28E-03 | FALSE |
| 550.2 | Diaphragmatic hernia | digestive | 1088 | 41,047 | 0.11 ± 0.03 | 1.12 (1.05–1.19) | 1.08E-03 | 5.53E-03 | FALSE |
| 379.3 | Aphakia and other disorders of lens | sense organs | 548 | 46,466 | 0.16 ± 0.05 | 1.17 (1.06–1.28) | 1.10E-03 | 5.53E-03 | FALSE |
| 367.2 | Astigmatism | sense organs | 683 | 47,064 | 0.14 ± 0.04 | 1.15 (1.06–1.25) | 1.25E-03 | 6.07E-03 | FALSE |
| 411.4 | Coronary atherosclerosis | circulatory system | 717 | 44,325 | 0.13 ± 0.04 | 1.14 (1.05–1.23) | 1.72E-03 | 8.04E-03 | FALSE |
| 591 | Urinary tract infection | genitourinary | 450 | 44,765 | 0.15 ± 0.05 | 1.16 (1.05–1.29) | 3.66E-03 | 1.65E-02 | FALSE |
| 535.8 | Other specified gastritis | digestive | 538 | 43,334 | 0.14 ± 0.05 | 1.15 (1.04–1.26) | 4.43E-03 | 1.93E-02 | FALSE |
| 480 | Pneumonia | respiratory | 251 | 46,963 | 0.19 ± 0.07 | 1.21 (1.05–1.38) | 6.77E-03 | 2.84E-02 | FALSE |
| 414 | Other forms of chronic heart disease | circulatory system | 293 | 44,325 | 0.17 ± 0.06 | 1.19 (1.05–1.34) | 7.20E-03 | 2.93E-02 | FALSE |
| 364.4 | Corneal degenerations | sense organs | 222 | 45,605 | 0.2 ± 0.07 | 1.22 (1.05–1.41) | 7.62E-03 | 3.00E-02 | FALSE |
| 394.7 | Disease of tricuspid valve | circulatory system | 200 | 47,125 | 0.2 ± 0.08 | 1.22 (1.05–1.43) | 9.20E-03 | 3.51E-02 | FALSE |
| 365 | Glaucoma | sense organs | 406 | 45,605 | 0.14 ± 0.05 | 1.15 (1.04–1.28) | 9.48E-03 | 3.51E-02 | FALSE |
| 561 | Symptoms involving digestive system | digestive | 575 | 36,758 | 0.12 ± 0.05 | 1.13 (1.03–1.23) | 9.81E-03 | 3.53E-02 | FALSE |
| 459.9 | Circulatory disease NEC | circulatory system | 943 | 45,329 | 0.09 ± 0.04 | 1.09 (1.02–1.17) | 1.26E-02 | 4.30E-02 | FALSE |
| 740.1 | Osteoarthritis; localized | musculoskeletal | 610 | 45,044 | 0.11 ± 0.05 | 1.12 (1.02–1.22) | 1.26E-02 | 4.30E-02 | FALSE |
| 428.2 | Heart failure NOS | circulatory system | 343 | 47,591 | 0.15 ± 0.06 | 1.16 (1.03–1.3) | 1.30E-02 | 4.31E-02 | FALSE |
| 530.1 | Reflux esophagitis | digestive | 399 | 41,524 | 0.13 ± 0.06 | 1.14 (1.03–1.27) | 1.50E-02 | 4.78E-02 | FALSE |
| 274.1 | Gout | endocrine/metabolic | 292 | 47,641 | 0.15 ± 0.06 | 1.17 (1.03–1.32) | 1.52E-02 | 4.78E-02 | FALSE |

BMFF was used as the predictor for each region. Multivariable logistic regression models were applied to test the associations between rank-transformed BMFF and each PheCODE, with adjustment for age at imaging, sex, and BMI at imaging. P values were corrected for multiple testing using the false discovery rate (FDR) Q value threshold of 0.05. Unadjusted P values (two-sided) and FDR-adjusted Q values are shown. Each PheCODE (disease outcome) was found to be significant after FDR correction (Q value < 0.05). Bonferroni = TRUE if P < 0.05 after Bonferroni adjustment for multiple comparisons.

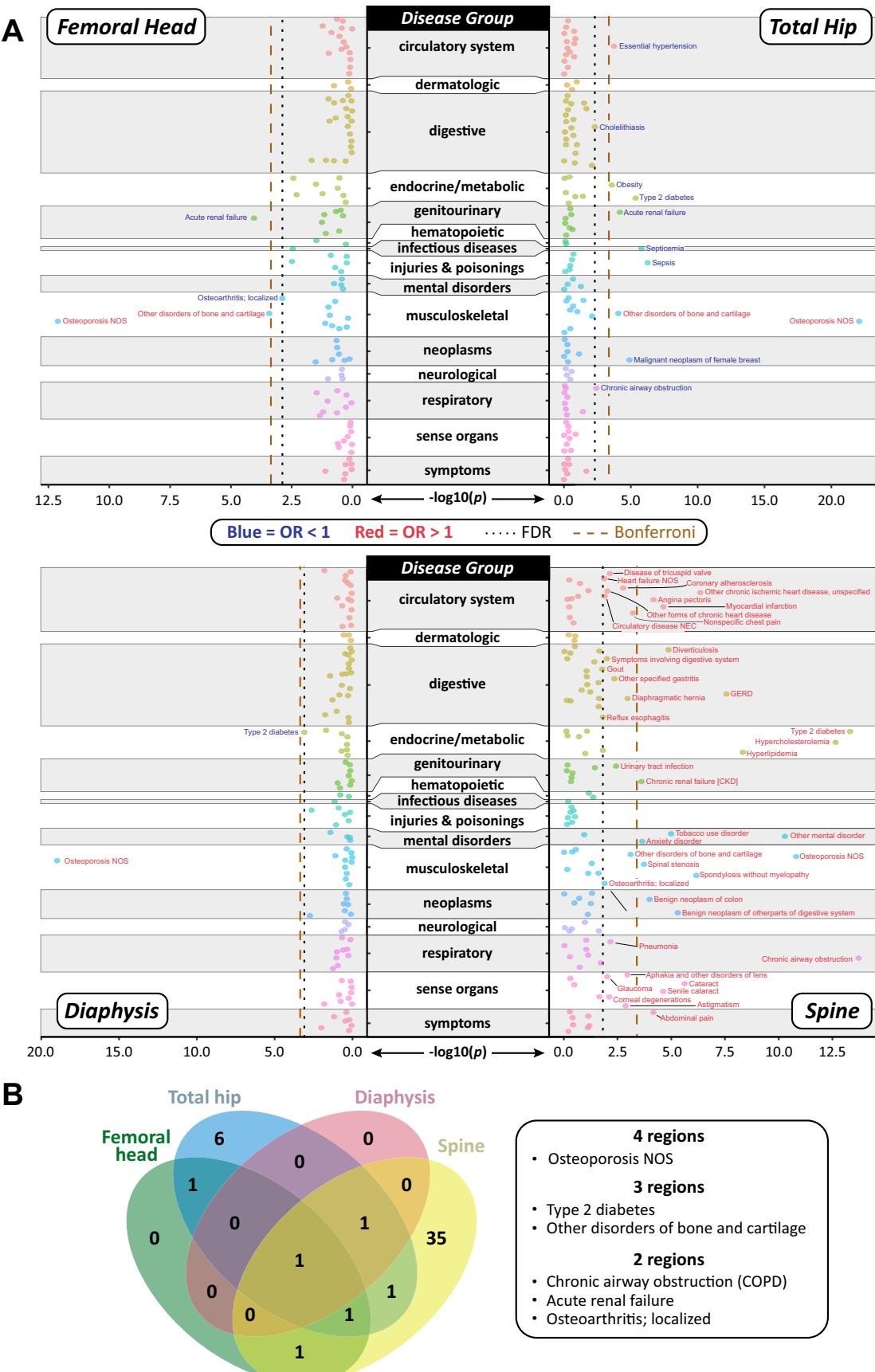

**Fig. 2 | Obs-PheWAS of associations between BMFF and incident diseases.** Multivariable logistic regression models were used to test the associations between rank-transformed BMFF and each PheCODE, with adjustment for age at imaging, sex, and BMI at imaging. P values were corrected for multiple testing using the false discovery rate (FDR) q value threshold of 0.05. **A** Manhattan plot for Femoral head, Total hip, Diaphysis, and Spine, showing −log10(P) values (two-sided) for each PheCODE. Labels of disease names are shown for all significant diseases (FDR < 0.05); those in blue font have an odds ratio (OR) < 1, while those in red font have OR > 1. Disease categories are shown on the vertical axis, with categories distinguished by alternating gray/white shading; the widths of these gray or white bars are proportional to the number of diseases analyzed within each category. The disease categories "Congenital abnormalities" and "Pregnancy complications" had zero incident cases and so are not shown. For each region, the dashed vertical lines indicate FDR (0.05) (dashed black line) or the Bonferroni-adjusted P value (dashed orange line). **B** Venn diagram showing significant disease associations shared between regions or unique to each region. The list beside the diagram shows the disease names shared for two, three, or all four regions. Further details are in Table 2 and Supplementary Data 2−3.

these, higher total hip BMFF was associated with increased risk of musculoskeletal diseases (osteoporosis and 'other disorders of bone and cartilage') but decreased risk of nine other diseases (T2D, essential hypertension, obesity, sepsis, septicemia, acute renal failure, cholelithiasis, chronic airway obstruction, and malignant neoplasm of female breast). It is helpful to note that "chronic airway obstruction" maps to ICD-10 codes J44.8/J44.9, for chronic obstructive pulmonary disease (COPD); hereafter, we refer to 'chronic airway obstruction' as COPD.

The diaphysis Obs-PheWAS identified two PheCODEs, increased risk of osteoporosis and decreased risk of T2D, that were associated with BMFF (Table 2; Fig. 2).

In the spine Obs-PheWAS, 40 PheCODEs were associated with higher BMFF. These could be categorized into ten disease groups and included diseases such as osteoporosis, 'other disorders of bone and cartilage', spondylosis without myelopathy, spinal stenosis, T2D, hypercholesterolemia, hyperlipidemia, gout, two benign neoplasms, and several cardiovascular conditions (Table 2; Fig. 2). The strongest effects were observed for COPD and osteoporosis, as well as metabolic diseases such as T2D and hyperlipidemia. Unlike for each femoral region, spine BMFF was also associated with numerous diseases affecting the circulatory system, digestive system, and sense organs (Table 2; Fig. 2). Moreover, whereas femoral BMFF was associated with decreased risk of some diseases, spine BMFF was associated with increased risk of all the identified disease groups.

Among all of the BMFF-associated PheCODES identified by Obs-PheWAS, six were shared between two or more bone regions (Fig. 2B; Supplementary Data 3). Osteoporosis was the one significant PheCODE with consistent direction of effect for all four bone regions, in each case showing a positive association with BMFF (Table 2; Supplementary Data 3). Another musculoskeletal PheCODE, 'other disorders of bone and cartilage', was positively associated with BMFF in the femoral head, total hip, and spine. T2D showed consistent negative associations with BMFF in the femoral head and total hip, but a positive association in the spine. Importantly, both spine BMFF and total hip BMFF remained significantly associated with T2D even after controlling for variation in visceral adipose tissue or overall peripheral adiposity, each of which is an independent risk factor for T2D[17,18] (Supplementary Data 4). Thus, these associations between BMFF and T2D are independent of adiposity outside of the bone marrow.

The three remaining PheCODEs shared for two bone regions were COPD, acute renal failure, and osteoarthritis. Intriguingly, each of these diseases was associated positively with spine BMFF but negatively with BMFF of the femoral head, total hip, and/or diaphysis (Table 2; Fig. 2; Supplementary Data 3). These findings support the concept that the functions and clinical implications of BMAT vary depending on the skeletal site.

For each bone region, we also conducted a sensitivity PheWAS analysis that included both incident and prevalent cases. This identified a total of 35, 43, 14, and 142 significant PheCODEs (FDR q < 0.05) associated with BMFF in the femoral head, total hip, femoral diaphysis, and spine, respectively (Supplementary Data 5; Supplementary Figs. 1–5). This analysis identified additional associations with fracture and site-specific associations with osteoarthritis, several cancers, chronic kidney disease, and various circulatory, digestive, genitourinary, and respiratory conditions. Forty-two PheCODEs were identified for two or more bone regions (Supplementary Data 6), and five PheCODEs showed significant associations for all four bone regions (osteoporosis, fracture of radius and ulna, T2D, gout, and essential hypertension) (Supplementary Fig. 1). The direction of effect for all overlapping diseases was consistent between the main Obs-PheWAS and the sensitivity Obs-PheWAS.

### Sex-sensitivity analysis for Obs-PheWAS

To investigate if any BMFF-associated diseases are sex-specific, we conducted the main (incident cases) and sensitivity Obs-PheWAS (incident + prevalent cases), stratified by sex, for the four bone regions. In males, one significant PheCODE was associated with BMFF in the femoral head, two in the total hip, none in the diaphysis, and 19 in the spine. In females, four, seven, three, and 11 significant PheCODEs were associated with BMFF in these respective bone regions (Supplementary Data 7).

The majority of identified significant diseases were also found in the sex-combined Obs-PheWAS (Table 2; Supplementary Data 7). In the spine, four PheCODEs, including T2D, hypercholesterolemia, other mental disorders, and gastroesophageal reflux disease (GERD), were positively associated with BMFF in both male and female groups. 'Essential hypertension' and 'iron deficiency anemia' were associated with spine BMFF in males, but not in females or in the sexes combined. All other significant PheCODEs identified for spine BMFF in the sex-sensitivity analysis were also observed in the sex-combined analysis (Table 2; Supplementary Data 7-8).

In contrast to spine BMFF, the incident diseases associated with BMFF at each femoral site were sex-specific, with no overlap between males and females (Supplementary Data 7). When prevalent cases were included, there was some male-female overlap among the diseases associated with each femoral site; for these diseases, the directions of the associations with BMFF were the same in each sex (Supplementary Data 9-10).

### Polygenic risk score (PRS)-PheWAS

**BMFF polygenic risk score.** To further explore the clinical implications of bone marrow adiposity, we generated PRSs for BMFF at each bone region and used these as the exposures for PRS-PheWAS across the larger UKBB cohort. We generated two PRSs, PRS1 and PRS2, for each bone region (Table 3). PRS1 included all SNPs that passed quality control (Supplementary Data 11). The same SNPs were used in PRS2, except that any SNPs independently associated with BMD were excluded (Supplementary Data 12). The rationale for this is that BMD-associated SNPs could confound the detection of associations between each BMFF PRS and osteoporosis, which was the disease most consistently associated with BMFF in our Obs-PheWAS. The only exception was for diaphysis, in which PRS1 contained no BMD-associated SNPs; hence, PRS2 was not required for this site. The quality control of genetic IVs (SNPs) and samples used to construct PRSs for each bone region is described in Supplementary Data 11–13, with the SNPs (rsid, chromosome, position, beta, $R^2$, F-statistics) used to generate each PRS described in Supplementary Data 14-15. The linear regression of PRS on BMFF in each bone region, stratified by sex, is

**Table 3 | Summary of the characteristics of the PRS samples for the four bone regions**

| | Femoral head (n = 305,313) | Total hip (n = 305,453) | Diaphysis (n = 306,222) | Spine (n = 303,066) |
|---|---|---|---|---|
| Age at assessment center (years)* | 57.04 (8.02) | 57.03 (8.02) | 57.04 (8.02) | 57.06 (8.02) |
| *Sex* | | | | |
| Male | 141,114 (46.22%) | 141,542 (46.34%) | 142,030 (46.38%) | 139,343 (45.98%) |
| Female | 164,199 (53.78%) | 163,911 (53.66%) | 164,192 (53.62%) | 163,723 (54.02%) |
| *Polygenic risk score* | | | | |
| PRS1 | 0.09 (0.34) | −0.16 (0.64) | −1.77 (0.75) | −0.53 (0.72) |
| PRS2 | 0.14 (0.31) | −0.11 (0.63) | NA | −0.47 (0.72) |

The polygenic risk score (PRS) for each bone region was constructed by adding up the number of BMFF-increasing alleles for each SNP, weighted for the SNP effect size on BMFF level (beta coefficients from meta-GWAS); a higher PRS value suggests a greater genetic predisposition to BMFF variation. The quality control of the SNPs and samples used to generate PRS for each bone region is summarized in Supplementary Data 11–13. The number of SNPs for PRS1 and PRS2 for each site are as follows: PRS1 – femoral head = 48, total hip = 115, diaphysis = 99, spine = 119; PRS2 – femoral head = 46, total hip = 114, spine = 117. For diaphysis, PRS2 was not generated because no SNPs in PRS1 (diaphysis) were independently associated with BMD. *Values are presented as mean (sd). Numbers of participants in each sex and ancestry category are shown as absolute number, with % of total for each bone region indicated in parentheses.

presented in Supplementary Fig. 6. Positive associations were observed, with $\beta$ coefficients (log OR) ranging from 0.189 to 0.381 ($P < 2 \times 10^{-16}$).

PRS-PheWAS was performed using PRS1 for each of the four bone regions, and PRS2 for femoral head, total hip, and spine, giving a total of seven separate PheWASes. The PheCODES used for PRS-PheWAS are summarized in Supplementary Data 16, while the baseline characteristics of each PRS-PheWAS sample for each bone region are summarized in Table 3. PheCODEs significantly associated with PRS1 or PRS2, after FDR correction, are presented in Table 4.

In the femoral head PRS-PheWAS, nine PheCODEs in four disease categories were associated with PRS1, including negative associations with two neoplastic diseases and positive associations with osteoporosis, fracture of humerus, and viral pneumonia (Table 4; Supplementary Fig. 7).

In the total hip PRS-PheWAS, five PheCODEs were associated with PRS1. Higher genetically predicted BMFF was associated with lower risk of myeloproliferative disease and 'other derangement of joint', but with higher risk of three musculoskeletal diseases: osteoporosis, pyogenic arthritis, and 'other disorders of bone and cartilage' (Table 4; Supplementary Fig. 7).

In the femoral diaphysis PRS-PheWAS, eight PheCODEs, spanning five disease groups, were associated with PRS1. Positive associations were observed with osteoporosis; multiple myeloma; esophagitis, GERD and related diseases; late effects of cerebrovascular disease; and fracture of the radius and ulna, hand or wrist, and femoral neck. In contrast, diaphysis PRS1 was negatively associated only with spinal stenosis (Table 4; Supplementary Fig. 7).

In the spine PRS-PheWAS, five PheCODEs demonstrated positive associations with PRS1, including osteoporosis and four PheCODEs for fractures at different skeletal sites; no PheCODEs were negatively associated with spine PRS1 (Table 4; Supplementary Fig. 7).

We found a high degree of overlap in significant PheCODEs identified across the two PRSs (Table 4; Fig. 3). PRS2 was associated with the same PheCODEs, with the same effect directions, as PRS1 for each site in the total hip, diaphysis, and spine; there were no major discrepancies in the association effect sizes, with a median OR difference of ~0.001. The only disparities for PRS1 vs PRS2 occurred for the femoral head, for which PRS1 was positively associated with both fracture of humerus and viral pneumonia, but PRS2 was not. Therefore, the BMD-associated SNPs, which were excluded from PRS2, have minimal influence on the PRS-PheWAS outcomes for each bone region.

Among all diseases identified by PRS1-PheWAS and PRS2-PheWAS, only three were common to two or more bone regions. Higher BMFF-PRS was associated with increased risk of osteoporosis for all four bone regions, with this relationship being stronger for each femoral site than for the spine (Supplementary Data 17). Fracture of radius and

ulna was positively associated with each PRS for the diaphysis and spine, whereas 'other derangement of joint' was inversely associated with each PRS for the femoral head and total hip (Supplementary Data 17). Together, these PRS-PheWAS analyses demonstrate that genetic markers of altered BMFF can be leveraged to further identify BMFF-associated diseases.

**Sex-sensitivity analysis for PRS-PheWAS.** In the sex-sensitivity PRS-PheWAS, no significant disease outcomes were identified as male-specific after FDR-q correction, and no diseases were associated with spine PRS1 or PRS2 in females (Supplementary Data 18). However, in females, several diseases were associated with PRS1 for the femoral head, total hip and diaphysis, and these overlapped entirely with the diseases associated with PRS2 for each of these sites (Supplementary Data 18). Among these, only osteoporosis was shared for all three femoral bone regions, for which higher BMFF PRS1 and PRS2 were consistently associated with the increased osteoporosis risk (Supplementary Data 19).

**Overlap significant PheCODEs in Obs-PheWAS (incident) and PRS-PheWAS.** Consistent positive associations with osteoporosis were found for BMFF in all four bone regions in both Obs-PheWAS and PRS-PheWAS. In addition, 'other disorders of bone and cartilage' were identified with increased disease risk for total hip BMFF (Supplementary Data 20).

The performance of BMFF-PRSs in predicting the risk of osteoporosis was evaluated across the four bone regions (Supplementary Data 21). The corrected C-statistics of PRS1 and PRS2 in each bone region were around 0.771, indicating a moderate-to-good discrimination of individuals with and without osteoporosis. There was no statistical difference in the predictive accuracy between PRS1 and PRS2 (ROC test $P > 0.05$).

**Mendelian randomization analyses.** We conducted two-sample MR to test if there is evidence of causality between altered BMFF and three of the diseases identified from our PheWAS analyses (Tables 5–7). Instrumental variables (IVs) were selected under two linkage disequilibrium (LD) clumping thresholds ($r^2 < 0.001$ or $r^2 < 0.6$) to evaluate robustness to variant correlation. Among all the PheCODEs identified from Obs-PheWAS, PRS1-PheWAS and PRS2-PheWAS, osteoporosis was the only disease consistently associated with all four bone regions (Supplementary Data 20). Fracture-related PheCODEs were also prominent among the PRS1- and PRS2-PheWAS results, while Obs-PheWAS revealed strong associations, in varying directions, between BMFF and T2D in total hip, diaphysis and spine (Table 2). Each of these diseases (osteoporosis, fractures, and T2D) has been a focus of previous BMAT research[2–4] and each imposes a substantial burden on

**Table 4 | Disease outcomes associated with the weighted polygenic risk score of BMFF in the PheWAS**

| Phe CODE | Description | Group | Cases (n) | Controls (n) | Beta ± SE | OR (95% CIs) | P value | Q value | Bonferroni |
|---|---|---|---|---|---|---|---|---|---|
| **Femoral head PRS1 (n = 9)** | | | | | | | | | |
| 743.11 | Osteoporosis NOS | musculoskeletal | 12,457 | 290,975 | 0.15 ± 0.03 | 1.16 (1.1–1.23) | 8.65E-08 | 8.78E-05 | TRUE |
| 742.9 | Other derangement of joint | musculoskeletal | 2806 | 297,082 | −0.27 ± 0.06 | 0.76 (0.69–0.85) | 1.27E-06 | 6.46E-04 | TRUE |
| 716.2 | Unspecified monoarthritis | musculoskeletal | 23,348 | 242,470 | −0.09 ± 0.02 | 0.91 (0.88–0.95) | 7.76E-06 | 2.62E-03 | TRUE |
| 742.8 | Articular cartilage disorder | musculoskeletal | 1057 | 297,082 | −0.35 ± 0.09 | 0.7 (0.59–0.84) | 8.35E-05 | 1.95E-02 | FALSE |
| 172.2 | Other non-epithelial cancer of skin | neoplasms | 26,248 | 274,134 | −0.08 ± 0.02 | 0.93 (0.89–0.96) | 9.60E-05 | 1.95E-02 | FALSE |
| 722.1 | Displacement of intervertebral disc | musculoskeletal | 618 | 281,750 | 0.46 ± 0.12 | 1.58 (1.25–2) | 1.49E-04 | 2.52E-02 | FALSE |
| 803.1 | Fracture of humerus | injuries & poisonings | 2566 | 276,850 | 0.22 ± 0.06 | 1.24 (1.11–1.4) | 2.21E-04 | 3.20E-02 | FALSE |
| 480.2 | Viral pneumonia | respiratory | 2100 | 281,426 | 0.24 ± 0.07 | 1.27 (1.11–1.44) | 2.93E-04 | 3.72E-02 | FALSE |
| 172.11 | Melanomas of skin | neoplasms | 5637 | 274,134 | −0.14 ± 0.04 | 0.87 (0.8–0.94) | 3.67E-04 | 4.14E-02 | FALSE |
| **Total hip PRS1 (n = 5)** | | | | | | | | | |
| 743.11 | Osteoporosis NOS | musculoskeletal | 12,423 | 291,149 | 0.12 ± 0.02 | 1.13 (1.1–1.17) | 2.76E-17 | 2.80E-14 | TRUE |
| 733 | Other disorders of bone and cartilage | musculoskeletal | 4785 | 281,306 | 0.1 ± 0.02 | 1.1 (1.05–1.15) | 2.91E-05 | 1.40E-02 | TRUE |
| 711.1 | Pyogenic arthritis | musculoskeletal | 524 | 242,631 | 0.28 ± 0.07 | 1.32 (1.16–1.52) | 4.65E-05 | 1.40E-02 | TRUE |
| 200 | Myeloproliferative disease | neoplasms | 1395 | 298,087 | −0.17 ± 0.04 | 0.85 (0.78–0.92) | 5.51E-05 | 1.40E-02 | FALSE |
| 742.9 | Other derangement of joint | musculoskeletal | 2821 | 297,206 | −0.11 ± 0.03 | 0.89 (0.84–0.95) | 1.16E-04 | 2.36E-02 | FALSE |
| **Diaphysis PRS1 (n = 8)** | | | | | | | | | |
| 743.11 | Osteoporosis NOS | musculoskeletal | 12,472 | 291,866 | 0.13 ± 0.01 | 1.13 (1.11–1.16) | 4.86E-24 | 4.93E-21 | TRUE |
| 803.2 | Fracture of radius and ulna | injuries & poisonings | 6701 | 277,713 | 0.09 ± 0.02 | 1.09 (1.06–1.13) | 1.85E-07 | 9.40E-05 | TRUE |
| 804 | Fracture of hand or wrist | injuries & poisonings | 4112 | 277,713 | 0.09 ± 0.02 | 1.1 (1.05–1.14) | 7.66E-06 | 2.59E-03 | TRUE |
| 204.4 | Multiple myeloma | neoplasms | 1068 | 298,828 | 0.17 ± 0.04 | 1.18 (1.09–1.28) | 5.10E-05 | 1.29E-02 | FALSE |
| 530.1 | Esophagitis, GERD and related diseases | digestive | 10,686 | 251,838 | 0.05 ± 0.01 | 1.05 (1.02–1.08) | 1.91E-04 | 3.87E-02 | FALSE |
| 720 | Spinal stenosis | musculoskeletal | 7029 | 282,615 | −0.06 ± 0.02 | 0.94 (0.91–0.97) | 2.30E-04 | 3.89E-02 | FALSE |
| 800.1 | Fracture of neck of femur | injuries & poisonings | 4208 | 277,713 | 0.08 ± 0.02 | 1.08 (1.04–112) | 2.88E-04 | 4.17E-02 | FALSE |
| 433.8 | Late effects of cerebrovascular disease | circulatory system | 2481 | 287,490 | 0.1 ± 0.03 | 1.1 (1.04–1.16) | 3.66E-04 | 4.64E-02 | FALSE |
| **Spine PRS1 (n = 5)** | | | | | | | | | |
| 803.3 | Fracture of clavicle or scapula | injuries & poisonings | 2685 | 274,721 | 0.1 ± 0.03 | 1.11 (1.05–1.17) | 1.18E-04 | 3.83E-02 | FALSE |
| 807 | Fracture of ribs | injuries & poisonings | 2726 | 274,721 | 0.07 ± 0.03 | 1.07 (1.02–2.13) | 9.58E-03 | 3.84E-02 | FALSE |
| 803.2 | Fracture of radius and ulna | injuries & poisonings | 6688 | 274,721 | 0.04 ± 0.02 | 1.04 (1.01–1.08) | 1.92E-02 | 3.89E-02 | FALSE |
| 800.3 | Fracture of tibia and fibula | injuries & poisonings | 2417 | 274,721 | 0.06 ± 0.03 | 1.07 (1.01–1.13) | 2.68E-02 | 4.84E-02 | FALSE |
| 743.11 | Osteoporosis NOS | musculoskeletal | 12,416 | 288,776 | 0.03 ± 0.01 | 1.03 (1–1.05) | 3.91E-02 | 3.05E-02 | FALSE |
| **Femoral head PRS2 (n = 7)** | | | | | | | | | |
| 742.9 | Other derangement of joint | musculoskeletal | 2806 | 297,082 | −0.31 ± 0.06 | 0.74 (0.65–0.83) | 6.75E-07 | 6.85E-04 | TRUE |
| 172.2 | Other non-epithelial cancer of skin | neoplasms | 26,248 | 274,134 | −0.09 ± 0.02 | 0.91 (0.87–0.95) | 1.06E-05 | 3.30E-03 | TRUE |
| 716.2 | Unspecified monoarthritis | musculoskeletal | 23,348 | 242,470 | −0.1 ± 0.02 | 0.91 (0.87–0.95) | 1.23E-05 | 3.30E-03 | TRUE |
| 743.11 | Osteoporosis NOS | musculoskeletal | 12,457 | 290,975 | 0.13 ± 0.03 | 1.14 (1.08–1.21) | 1.30E-05 | 3.30E-03 | TRUE |
| 742.8 | Articular cartilage disorder | musculoskeletal | 1057 | 297,082 | −0.41 ± 0.1 | 0.66 (0.54–0.8) | 3.39E-05 | 6.89E-03 | TRUE |
| 172.11 | Melanomas of skin | neoplasms | 5637 | 274,134 | −0.17 ± 0.04 | 0.85 (0.78–0.92) | 1.47E-04 | 2.49E-02 | FALSE |
| 722.1 | Displacement of intervertebral disc | musculoskeletal | 618 | 281,750 | 0.48 ± 0.13 | 1.61 (1.25–2.08) | 2.67E-04 | 3.88E-02 | FALSE |
| **Total hip PRS2 (n = 5)** | | | | | | | | | |

**Table 4 (continued) | Disease outcomes associated with the weighted polygenic risk score of BMFF in the PheWAS**

| Phe CODE | Description | Group | Cases (n) | Controls (n) | Beta ± SE | OR (95% CIs) | P value | Q value | Bonferroni |
|---|---|---|---|---|---|---|---|---|---|
| 743.11 | Osteoporosis NOS | musculoskeletal | 12,423 | 291,149 | 0.12 ± 0.02 | 1.13 (1.1–1.16) | 2.16E-16 | 2.19E-13 | TRUE |
| 733 | Other disorders of bone and cartilage | musculoskeletal | 4785 | 281,306 | 0.1 ± 0.02 | 1.1 (1.05–1.15) | 2.88E-05 | 1.30E-02 | TRUE |
| 711.1 | Pyogenic arthritis | musculoskeletal | 524 | 242,631 | 0.29 ± 0.07 | 1.33 (1.16–1.52) | 4.55E-05 | 1.30E-02 | TRUE |
| 200 | Myeloproliferative disease | neoplasms | 1395 | 298,087 | −0.17 ± 0.04 | 0.84 (0.78–0.92) | 5.12E-05 | 1.30E-02 | FALSE |
| 742.9 | Other derangement of joint | musculoskeletal | 2821 | 297,206 | −0.12 ± 0.03 | 0.89 (0.84–0.94) | 1.03E-04 | 2.09E-02 | FALSE |
| **Spine PRS2 (n = 5)** | | | | | | | | | |
| 803.3 | Fracture of clavicle or scapula | injuries & poisonings | 2685 | 274,721 | 0.1 ± 0.03 | 1.11 (1.05–1.17) | 1.47E-04 | 3.83E-02 | FALSE |
| 807 | Fracture of ribs | injuries & poisonings | 2726 | 274,721 | 0.07 ± 0.03 | 1.07 (1.02–1.13) | 1.13E-02 | 3.82E-02 | FALSE |
| 800.3 | Fracture of tibia and fibula | injuries & poisonings | 2417 | 274,721 | 0.06 ± 0.03 | 1.06 (1.01–1.12) | 3.35E-02 | 4.80E-02 | FALSE |
| 803.2 | Fracture of radius and ulna | injuries & poisonings | 6688 | 274,721 | 0.04 ± 0.02 | 1.04 (1–1.07) | 3.55E-02 | 3.05E-02 | FALSE |
| 743.11 | Osteoporosis NOS | musculoskeletal | 12,416 | 288,776 | 0.02 ± 0.01 | 1.02 (1–1.05) | 3.82E-02 | 3.19E-02 | FALSE |

For each region, PRS1 or PRS2 was used as the predictor, as indicated. The number of SNPs in PRS1 or PRS2 is as described for Table 3. Multivariable logistic regression models were applied to test the associations between BMFF PRS and each PheCODE, adjusting for age, sex, UK Biobank assessment center, and the first 10 genetic principal components (PC). P values were corrected for multiple testing using the false discovery rate (FDR) Q value threshold of 0.05. Unadjusted P values (two-sided) and FDR-adjusted Q values are shown. Each PheCODE (disease outcome) was found to be significant after FDR correction (Q value < 0.05). Bonferroni = TRUE if P < 0.05 after Bonferroni adjustment for multiple comparisons.

public health. Therefore, we focused our MR analyses on these three diseases.

In the MR analysis, the proportion of variance ($R^2$) explained by the genetic variants ranged from 0.06% to 0.41% and the F-statistic ranged from 29.82 to 179.79 across the four bone regions, indicating that genetically predicted BMFF is a robust IV for the MR analysis (Supplementary Data 15, 22, 23 and 24).

The causal association of BMFF genetic predisposition with osteoporosis is summarized in Table 5. When using the more stringent LD clumping threshold ($r^2 < 0.001$), positive associations with osteoporosis were found for total hip BMFF and diaphysis BMFF based on the inverse-variance weighted (IVW) method and for diaphysis BMFF based on MR-Horse. Moreover, consistent positive associations were detected for all three femoral sites when the less-stringent LD clumping threshold of $r^2 < 0.6$ was applied (Supplementary Data 25). However, no causal association was found in the spine, regardless of the $r^2$ threshold used. As outlined in the Discussion, the lack of causality for spine BMFF is surprising and could have several explanations, including osteoporosis diagnoses in UKBB being driven more by bone loss in the hip than in the spine. The Cochran's Q statistic of MR-Egger and IVW methods indicated heterogeneity (Table 5, Supplementary Data 25). However, MR-Egger intercept in the analysis showed no horizontal pleiotropy for each bone region. The scatter plots, forest plots and funnel plots for these analyses are presented in Supplementary Figs. 8–23. The results were also consistent after leave-one-out analysis (Supplementary Figs. 10, 14, 18, 22; Supplementary Data 22, 26). The association of osteoporosis remained the same in the MR-PRESSO analyses (Supplementary Data 27) after removal of outliers, for the femoral sites. There was no difference in MR-PRESSO outlier correction-adjusted causal estimates. The MR-PRESSO global test indicated evidence of heterogeneity across all regions. Distortion coefficients for femoral head, total hip, and diaphysis were non-significant, suggesting more robust estimates after outlier correction.

The causal association of BMFF genetic liability with fractures is summarized in Table 6. Each of the MR methods found no significant causal associations for any of the four bone regions when using the more stringent LD clumping threshold ($r^2 < 0.001$). However, when the more lenient LD clumping threshold of $r^2 < 0.6$ was applied, IVW and MR-Horse showed positive associations with fractures for all three femoral sites (Supplementary Data 28). Horizontal pleiotropy was found only in MR for the femoral head ($P = 0.015$) when using the more lenient LD clumping threshold. The scatter plots, forest plots and funnel plots are presented in Supplementary Figs. 24–39. The results were consistent after leave-one-out analysis (Supplementary Figs. 26, 30, 34, 38; Supplementary Data 23, 29). MR-PRESSO analyses (Supplementary Data 27) also supported the positive causal associations with fractures in femoral head and total hip. MR-PRESSO global test found evidence of heterogeneity only for the diaphysis when using the more lenient LD clumping threshold.

The causal association of BMFF genetic liability with T2D is summarized in Table 7. When using the more stringent LD clumping threshold ($r^2 < 0.001$), none of the MR methods found significant causal associations in any of the four bone regions. At the less-stringent LD clumping threshold ($r^2 < 0.6$), the findings suggested negative associations with T2D in the total hip and diaphysis, whereas MR-Horse observed a positive association between BMFF genetic liability and T2D in the spine (Supplementary Data 30). Evidence of heterogeneity was detected across all regions, but pleiotropy (Egger intercept) was not significant in most cases (only P-diaphysis = 0.022 for MR using $r^2 < 0.6$), supporting the validity of the causal estimates. The scatter plots, forest plots and funnel plots are presented in Supplementary Figs. 40–55. The results remained consistent in leave-one-out analysis (Supplementary Figs. 42, 46, 50, 54; Supplementary Data 24, 31). The MR-PRESSO global test (Supplementary Data 27) indicated evidence of heterogeneity across all regions. Only the

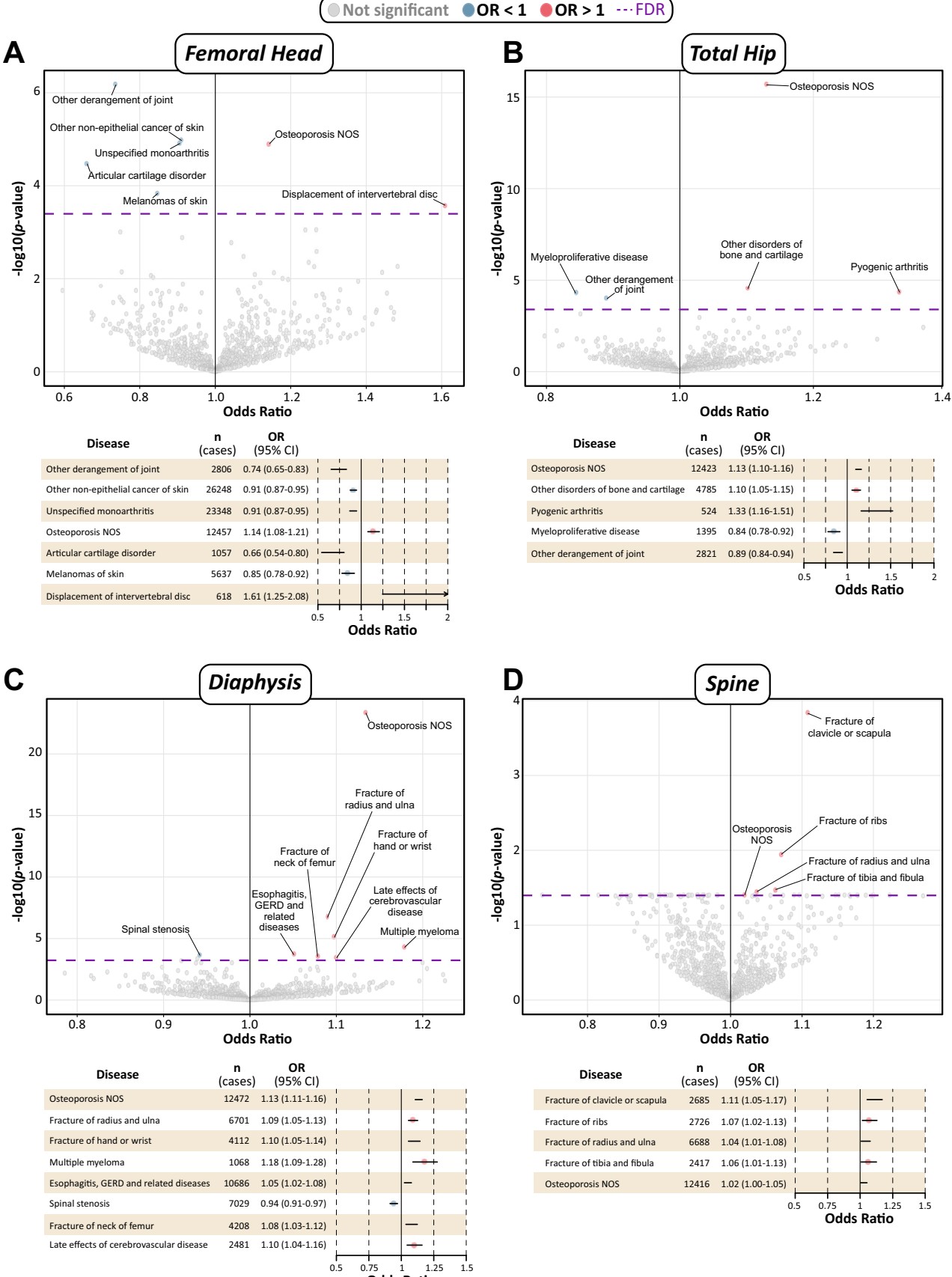

**Fig. 3 | PRS-PheWAS of associations between BMFF polygenic risk score (PRS) and diseases.** Multivariable logistic regression models were used to test the associations between BMFF PRS and each PheCODE, adjusting for age, sex, assessment center, and the first 10 genetic principal components. We applied the FDR correction ($q$ value < 0.05 as statistically significant) following the method by Benjamini–Hochberg to account for multiple testing in each of the phenotype-wide analyses. **A–D** Volcano plots show odds ratios (OR) and −log10(P) values for Phe-CODE diseases associated with femoral head PRS2 (**A**), total hip PRS2 (**B**), diaphysis PRS1 (**C**), and spine PRS2 (**D**). For PRS2, each PRS was first calculated based on significant SNPs from meta-GWAS after QC; SNPs independently associated with

BMD were then excluded from the instrumental variants selected. PRS2 for diaphysis was not calculated due to the absence of a corresponding BMD GWAS. Each dot represents a disease, with the $x$ axis indicating the OR and the $y$ axis showing the −log10(p) (two-sided). Labels of disease names are shown for all significant diseases (FDR < 0.05); significant associations with OR < 1 are shown as blue dots; those with OR > 1 are red dots; and non-significant associations are shown as gray dots. For each region, the dashed horizontal line indicates the FDR threshold. Forest plots beneath each volcano plot are included to show the OR (red or blue circles) with 95% CIs. Further details are in Table 4 and Supplementary Data 16, 17. Volcano plots for PRS1-PheWAS are presented in Supplementary Fig. 7.

diaphysis region found a significant distortion coefficient under all $r^2$ thresholds, indicating potential influence from outliers. Overall, the association of T2D remained consistent in the MR-PRESSO analyses after removal of outliers.

Finally, to assess whether BMI adjustment introduced collider bias in the MR, we compared MR results ($r^2 < 0.001$) based on IVs from meta-GWAS with and without BMI adjustment[15]. We found no distinct discrepancies in the comparisons (Supplementary Data 32–34), with the effect estimates remaining directionally consistent between BMI-adjusted and unadjusted analyses (Osteoporosis: median OR difference = 0.027; Fractures: median OR difference = 0.022; T2D: median OR difference = 0.004). Thus, BMI adjustment does not significantly alter the MR results.

## Discussion

Our understanding of the diseases associated with altered bone marrow adiposity has been based on smaller-scale studies incapable of comprehensively revealing the pathophysiological relevance of BMAT. These have typically used case-control designs addressing specific pathophysiological contexts; have lacked comprehensive healthcare records and genotype data for participants; often focused on only one or two skeletal sites; and have never exceeded more than 729 individuals[13]. Herein, we have overcome these limitations by using our deep-learning measurements of BMFF at the femoral head, total hip, femoral diaphysis, and spine of over 44,000 participants in the UKBB, as well as establishing PRSs for BMFF at each of these sites. Our Obs-PheWAS provides a systematic identification of the clinical conditions associated with altered BMFF at each site, while our PRS-PheWAS demonstrates the utility of genetic markers of altered BMFF for further elucidating clinical associations. These analyses greatly extend the scope of diseases associated with altered bone marrow adiposity. Finally, our MR studies reveal that genetic predisposition to increased BMFF at the total hip and femoral diaphysis is causally associated with osteoporosis and suggest that spine and femoral BMFF may also be causally associated with fractures and T2D. Together, these findings substantially advance understanding of the impact of BMAT on human health and disease.

The entire scope of BMFF-associated diseases is too extensive to discuss here in full; hence, below we consider those of particular interest.

Across all the BMFF-associated diseases, the relationship with osteoporosis (PheCODE 743.11) is among the strongest and most consistent. This PheCODE includes ICD-9 code 733.00 (Osteoporosis NOS) and ICD-10 codes M81, M81.0, M81.6 and M81.8, which refer to age-related, localized, or "other" osteoporosis, without current pathological fractures. From our incident Obs-PheWAS, PRS1- and PRS2-PheWAS, osteoporosis is the only disease associated with BMFF at all four sites. This extends numerous reports of the inverse relationship between bone marrow adiposity and BMD[3], including our recent population-scale findings in the UKBB[15]. The persistence of this relationship in our PRS2-PheWAS is intriguing because PRS2 excludes

BMD-associated SNPs, demonstrating that the relationship between genetically predicted BMFF and osteoporosis is independent of genetic markers of altered BMD.

Our PheWAS and MR findings confirm and greatly extend previous observations, as well as strengthen the evidence of causality between increased BMFF and osteoporosis in humans. Intriguingly, the observed causal association exists for BMFF at each femoral site, particularly when using the more lenient LD threshold, but not for spine BMFF. This is consistent with our PheWAS findings, which show weaker BMFF-osteoporosis relationships for the spine than for each femoral site; however, it is surprising given our finding that spine BMFF has a robust inverse association with spine BMD[15]. This discordance could result from an indirect relationship between BMFF and BMD in the spine: either increased spine BMFF is a consequence, rather than a cause, of lower BMD, or these are independent effects driven simultaneously by other factors. However, another possibility is that osteoporosis diagnoses in UKBB are driven more by decreased BMD in the hip than in the spine. Indeed, BMD T-scores across the UKBB population are lower in the proximal femur than in the spine (see UKBB data fields 23300 vs 23205, respectively), echoing observations in other aged populations[19]. If low spine BMD plays a lesser role in UKBB osteoporosis cases, this may explain our finding that increased spine BMFF has no causal association with osteoporosis, despite its robust association with decreased spine BMD.

Among the femoral sites, the causal association with osteoporosis is strongest for diaphysis BMFF. The biological reasons for this are unclear but may result from site-specific differences in BMAT formation and function[2,20,21]. For example, bone marrow adipocytes secrete a range of anti-osteogenic and/or pro-resorptive factors[2], and this may differ between the diaphysis and the proximal femur. Another theory is that, because BM adipocytes and osteoblasts derive from a common progenitor (the skeletal stem cell), any increase in adipogenesis will necessarily come at the expense of osteogenesis[2]. Indeed, site-specific differences in human skeletal stem cells have recently been identified[22]. Thus, another possibility is that the balance between adipogenesis and osteoblastogenesis varies across skeletal sites, contributing to site-specific differences in the BMFF-osteoporosis relationship. Notably, relatively few studies measure bone marrow adiposity in the femoral diaphysis, instead focusing on the spine and proximal femur[3]. These site-specific differences may also help to inform the potential of using bone marrow adiposity to improve clinical management of osteoporosis.

Another key finding is that genetically predicted BMFF is associated with fractures, with MR providing evidence for causality when using the more lenient LD threshold. This is important because fractures impose a huge burden on public health[23] and, although BMD is a major determinant of fracture risk, most fractures occur in people without osteoporosis[24]. This highlights the need for improved fracture prediction, which has motivated research into bone marrow adiposity as a fracture risk factor[3]. Previous studies show that increased vertebral bone marrow adiposity is associated with prevalent vertebral fracture in men, independently of BMD[25], and with incident bone loss in elderly women[26]. However, others report that bone marrow adiposity in

**Table 5 | Associations of genetically predicted BMFF levels with osteoporosis in Mendelian randomization**

| BMFF-osteoporosis MR | | | | | Heterogeneity | | | Pleiotropy | | |
|---|---|---|---|---|---|---|---|---|---|---|
| Method | Beta ± SE | OR (95% CIs) | *P* value | FDR-Q | *Q* | Q_df | *Q_pval* | Egger intercept | SE | *pval* |
| **Femoral head** | | | | | | | | | | |
| MR Horse | 0.435 ± 0.267 | 1.545 (0.937–2.630) | 1.03E-01 | 2.07E-01 | - | - | - | - | - | - |
| IVW | 0.222 ± 0.225 | 1.249 (0.803–1.941) | 3.24E-01 | 4.09E-01 | 73.871 | 14 | 3.81E-10 | - | - | - |
| MR Egger | 1.929 ± 0.823 | 6.884 (1.373-34.516) | 3.60E-02 | 9.50E-02 | 54.623 | 13 | 4.71E-07 | −0.088 | 0.041 | 5.19E-02 |
| Weighted median | 0.070 ± 0.175 | 1.073 (0.762–1.511) | 6.88E-01 | 7.86E-01 | - | - | - | - | - | - |
| Simple mode | 0.006 ± 0.332 | 1.006 (0.525–1.928) | 9.86E-01 | 9.86E-01 | - | - | - | - | - | - |
| Weighted mode | −0.007 ± 0.366 | 0.993 (0.484–2.034) | 9.84E-01 | 9.86E-01 | - | - | - | - | - | - |
| **Total hip** | | | | | | | | | | |
| MR Horse | 0.180 ± 0.144 | 1.197 (0.902–1.595) | 2.11E-01 | 3.05E-01 | - | - | - | - | - | - |
| IVW | 0.559 ± 0.171 | 1.749 (1.251–2.447) | **1.00E-03** | **5.00E-03** | 179.900 | 27 | 1.55E-24 | - | - | - |
| MR Egger | 1.253 ± 0.497 | 3.501 (1.323–9.266) | **1.80E-02** | 6.20E-02 | 165.855 | 26 | 2.49E-22 | −0.041 | 0.028 | 1.50E-01 |
| Weighted median | 0.208 ± 0.121 | 1.231 (0.972–1.560) | 8.50E-02 | 1.86E-01 | - | - | - | - | - | - |
| Simple mode | −0.110 ± 0.164 | 0.895 (0.649–1.235) | 5.07E-01 | 6.08E-01 | - | - | - | - | - | - |
| Weighted mode | −0.009 ± 0.177 | 0.991 (0.700–1.402) | 9.58E-01 | 9.86E-01 | - | - | - | - | - | - |
| **Diaphysis** | | | | | | | | | | |
| MR Horse | 0.775 ± 0.152 | 2.171 (1.624–2.936) | **3.42E-07** | **2.74E-06** | - | - | - | - | - | - |
| IVW | 0.841 ± 0.159 | 2.320 (1.698–3.168) | **1.23E-07** | **1.47E-06** | 101.399 | 18 | 1.23E-13 | - | - | - |
| MR Egger | 1.221 ± 0.494 | 3.390 (1.287–8.926) | **2.40E-02** | 7.30E-02 | 97.612 | 17 | 2.46E-13 | −0.025 | 0.030 | 4.28E-01 |
| Weighted median | 0.914 ± 0.127 | 2.495 (1.944–3.202) | **6.71E-13** | **1.61E-11** | - | - | - | - | - | - |
| Simple mode | 0.812 ± 0.221 | 2.253 (1.461–3.475) | **2.00E-03** | **7.00E-03** | - | - | - | - | - | - |
| Weighted mode | 0.898 ± 0.221 | 2.454 (1.591–3.786) | **1.00E-03** | **4.00E-03** | - | - | - | - | - | - |
| **Spine** | | | | | | | | | | |
| MR Horse | −0.110 ± 0.080 | 0.896 (0.765–1.053) | 1.69E-01 | 2.71E-01 | - | - | - | - | - | - |
| IVW | −0.123 ± 0.082 | 0.884 (0.754–1.038) | 1.31E-01 | 2.43E-01 | 57.513 | 33 | 5.17E-03 | - | - | - |
| MR Egger | −0.615 ± 0.303 | 0.541 (0.299–0.979) | 5.10E-02 | 1.21E-01 | 52.833 | 32 | 1.17E-02 | 0.026 | 0.016 | 1.02E-01 |
| Weighted median | −0.141 ± 0.103 | 0.868 (0.710–1.062) | 1.69E-01 | 2.71E-01 | - | - | - | - | - | - |
| Simple mode | −0.288 ± 0.287 | 0.750 (0.427–1.316) | 3.23E-01 | 4.09E-01 | - | - | - | - | - | - |
| Weighted mode | −0.350 ± 0.278 | 0.705 (0.409–1.215) | 2.16E-01 | 3.05E-01 | - | - | - | - | - | - |

Exposure: BMFF (PMID: 39747859); Outcome: Osteoporosis (GCST90038656). Number of SNPs for clump $r^2 < 0.001$: femoral head=15; total hip=28; diaphysis=19; spine=34. Heterogeneity was assessed by the inverse-variance weighted (IVW) method and by MR Egger. Pleiotropy was assessed from MR Egger. All tests were two-sided. MR applied the FDR-Q correction (Q value < 0.05 as statistically significant).
*P* values and FDR-Q values < 0.05 are shown in bold.

vertebrae and the proximal femur does not differ between subjects with and without prevalent fracture[27–29]. Moreover, the largest previous study of bone marrow adiposity and incident fracture included 298 men and women but found no association between vertebral bone marrow adiposity and incident fractures in either sex[26]. This is similar to our main Obs-PheWAS results, in which there are no associations between BMFF and incident fractures, regardless of BMFF site or fracture type. However, the lack of associations with incident fracture is likely explained by the low fracture rates in the UKBB. Indeed, among all incident fractures, even the most common subtype (fracture of the radius and ulna) has <200 incident cases (Supplementary Data 35), which is below the threshold for inclusion as a PheWAS outcome. This threshold is exceeded for many fracture subtypes in our sensitivity PheWASes (Supplementary Data 36), which show robust positive associations between fractures and BMFF at each site; for spine and diaphysis, this is replicated in our PRS-PheWAS. The associations with fracture in our sensitivity Obs-PheWAS and PRS-PheWAS may also be due to reverse causality; hence, in the future, it will be important to conduct even larger studies to increase the power to detect associations between BMFF and incident fractures. Nevertheless, our findings highlight roles for diaphyseal BMFF, which previous studies have overlooked, and show that genetically predicted BMFF is positively associated with fracture, independent of genetic predictors of BMD.

As for osteoporosis, our MR results for fracture represent a major advance in knowledge because they demonstrate evidence of causality. These causal associations are stronger for each femoral site than for spine BMFF, echoing our MR findings for osteoporosis. Indeed, it is intriguing that spine BMFF may directly affect fractures but not osteoporosis. This underscores the concept that factors influencing bone quality, beyond BMD, contribute to fracture pathogenesis[30,31]. However, causality was not observed using the more stringent MR LD threshold ($r^2 < 0.001$), highlighting the need for stronger IVs to test causal associations. It will also be important for future studies to establish how increased BMFF at each site impacts bone quality and thereby contributes to fracture risk, and if bone marrow adiposity, and/or its genetic surrogates, can improve fracture prediction beyond traditional risk factors.

In addition to musculoskeletal diseases, the relationship between bone marrow adiposity and T2D has attracted increasing attention[4]. Several previous studies report increased vertebral bone marrow adiposity in T2D, including obese subjects[32,33], non-obese women[34], and older men[35]. In contrast, a recent larger study found only a trend for increased vertebral bone marrow adiposity[36], while earlier research detected no differences between control and T2D subjects[27,37,38]. However, the largest of these studies included only 199 people[36], and most have focused only on bone marrow adiposity in the spine,

**Table 6 | Associations of genetically predicted BMFF levels with fractures in Mendelian randomization**

| BMFF-fractures MR | | | | | Heterogeneity | | | Pleiotropy | | |
|---|---|---|---|---|---|---|---|---|---|---|
| Method | Beta ± SE | OR (95% CIs) | P value | FDR-Q | Q | Q_df | Q_pval | Egger intercept | SE | pval |
| **Femoral head** | | | | | | | | | | |
| MR Horse | 0.188 ± 0.109 | 1.207 (0.970–1.496) | 8.50E-02 | 2.81E-01 | - | - | - | - | - | - |
| IVW | 0.214 ± 0.101 | 1.239 (1.017–1.510) | **3.30E-02** | 2.81E-01 | 17.153 | 14 | 2.48E-01 | - | - | - |
| MR Egger | 0.705 ± 0.405 | 2.023 (0.915–4.472) | 1.05E-01 | 2.81E-01 | 15.315 | 13 | 2.88E-01 | −0.025 | 0.020 | 2.34E-01 |
| Weighted median | 0.252 ± 0.130 | 1.286 (0.996–1.660) | 5.30E-02 | 2.81E-01 | - | - | - | - | - | - |
| Simple mode | 0.303 ± 0.236 | 1.354 (0.853–2.149) | 2.19E-01 | 3.45E-01 | - | - | - | - | - | - |
| Weighted mode | 0.308 ± 0.245 | 1.361 (0.843–2.198) | 2.28E-01 | 3.45E-01 | - | - | - | - | - | - |
| **Total hip** | | | | | | | | | | |
| MR Horse | 0.114 ± 0.067 | 1.121 (0.979–1.284) | 8.90E-02 | 2.81E-01 | - | - | - | - | - | - |
| IVW | 0.106 ± 0.062 | 1.112 (0.985–1.256) | 8.70E-02 | 2.81E-01 | 27.495 | 27 | 4.37E-01 | - | - | - |
| MR Egger | 0.233 ± 0.186 | 1.262 (0.876–1.817) | 2.22E-01 | 3.45E-01 | 26.957 | | 4.12E-01 | −0.007 | 0.010 | 4.77E-01 |
| Weighted median | 0.200 ± 0.089 | 1.221 (1.026–1.453) | 2.50E-02 | 2.81E-01 | - | - | - | - | - | - |
| Simple mode | 0.185 ± 0.150 | 1.203 (0.896–1.615) | 2.29E-01 | 3.45E-01 | - | - | - | - | - | - |
| Weighted mode | 0.218 ± 0.126 | 1.243 (0.971–1.592) | 9.60E-02 | 2.81E-01 | - | - | - | - | - | - |
| **Diaphysis** | | | | | | | | | | |
| MR Horse | 0.086 ± 0.079 | 1.090 (0.937–1.273) | 2.76E-01 | 3.90E-01 | - | - | - | - | - | - |
| IVW | 0.111 ± 0.077 | 1.118 (0.961–1.300) | 1.48E-01 | 3.45E-01 | 27.544 | 18 | 6.93E-02 | - | - | - |
| MR Egger | 0.398 ± 0.232 | 1.489 (0.944–2.348) | 1.05E-01 | 2.81E-01 | 25.042 | 17 | 9.38E-02 | −0.019 | 0.014 | 2.10E-01 |
| Weighted median | 0.127 ± 0.094 | 1.136 (0.945–1.366) | 1.76E-01 | 3.45E-01 | - | - | - | - | - | - |
| Simple mode | 0.101 ± 0.159 | 1.106 (0.810–1.509) | 5.35E-01 | 7.13E-01 | - | - | - | - | - | - |
| Weighted mode | 0.151 ± 0.122 | 1.163 (0.916–1.477) | 2.30E-01 | 3.45E-01 | - | - | - | - | - | - |
| **Spine** | | | | | | | | | | |
| MR Horse | −0.008 ± 0.061 | 0.992 (0.882–1.121) | 8.96E-01 | 9.10E-01 | - | - | - | - | - | - |
| IVW | −0.020 ± 0.057 | 0.980 (0.876–1.097) | 7.31E-01 | 8.77E-01 | 28.381 | 33 | 6.96E-01 | - | - | - |
| MR Egger | −0.028 ± 0.219 | 0.972 (0.633–1.493) | 8.98E-01 | 9.10E-01 | 28.380 | 32 | 6.50E-01 | 0.0005 | 0.011 | 9.68E-01 |
| Weighted median | 0.009 ± 0.082 | 1.009 (0.859–1.186) | 9.10E-01 | 9.10E-01 | - | - | - | - | - | - |
| Simple mode | −0.052 ± 0.187 | 0.950 (0.658–1.371) | 7.85E-01 | 8.97E-01 | - | - | - | - | - | - |
| Weighted mode | 0.076 ± 0.177 | 1.079 (0.762–1.527) | 6.70E-01 | 8.47E-01 | - | - | - | - | - | - |

Exposure: BMFF (PMID: 39747859); Outcome: Fractures (GCST90038703). Number of SNPs for clump $r^2 < 0.001$: femoral head=15; total hip=28; diaphysis=19; spine=34. Heterogeneity was assessed by the inverse-variance weighted (IVW) method and by MR Egger. Pleiotropy was assessed from MR Egger. All tests were two-sided. MR applied the FDR-Q correction (Q value < 0.05 as statistically significant).
P values < 0.05 are shown in bold.

overlooking appendicular bones. Our present observations, therefore, substantially advance understanding of this relationship through large-scale analyses of multiple skeletal sites.

Perhaps our most intriguing finding is that, unlike osteoporosis and fracture, the relationship with T2D is positive for spine BMFF but negative for each femoral site, particularly for the total hip. This divergence between the spine and femur occurs for several other T2D-associated diseases, including obesity and gout[39]. It is also consistent with recent research from Badr et al., which found that, unlike for vertebrae, proximal femoral bone marrow adiposity is lower in T2D patients vs controls[36]. One possibility is that these BMFF-T2D associations are confounded by common associations with peripheral adiposity traits. Indeed, T2D is strongly associated with adipose distribution, with excessive visceral adiposity being a major risk factor for this disease[17,18]. Moreover, we previously demonstrated that spine BMFF is positively associated with total body fat % and visceral adiposity, whereas total hip and diaphysis BMFF are each negatively associated with visceral adiposity and have variable relationships with other adiposity traits[15]. However, we show that the BMFF-T2D associations for spine and total hip BMFF persist even after controlling for peripheral adiposity. Thus, BMFF at these sites is significantly associated with T2D incidence, independently of adiposity outside of the bone marrow.

**Table 7 | Associations of genetically predicted BMFF levels with Type 2 diabetes in Mendelian randomization**

| BMFF-T2D MR | | | | | Heterogeneity | | | Pleiotropy | | |
|---|---|---|---|---|---|---|---|---|---|---|
| Method | Beta ± SE | OR (95% CIs) | P value | FDR-Q | Q | Q_df | Q_pval | Egger intercept | SE | pval |
| **Femoral head** | | | | | | | | | | |
| MR Horse | 0.072 ± 0.054 | 0.931 (0.968–1.197) | 1.82E-01 | 3.97E-01 | - | - | - | - | - | - |
| IVW | 0.062 ± 0.056 | 1.064 (0.953–1.189) | 2.69E-01 | 4.97E-01 | 30.651 | 14 | 6.20E-03 | - | - | - |
| MR Egger | −0.014 ± 0.245 | 0.986 (0.610–1.592) | 9.54E-01 | 9.78E-01 | 30.406 | 13 | 4.11E-03 | 0.004 | 0.013 | 7.51E-01 |
| Weighted median | 0.094 ± 0.064 | 1.098 (0.969–1.244) | 1.42E-01 | 3.41E-01 | - | - | - | - | - | - |
| Simple mode | 0.111 ± 0.113 | 1.118 (0.895–1.396) | 3.43E-01 | 5.49E-01 | - | - | - | - | - | - |
| Weighted mode | 0.119 ± 0.120 | 1.126 (0.890–1.426) | 3.39E-01 | 5.49E-01 | - | - | - | - | - | - |
| **Total hip** | | | | | | | | | | |
| MR Horse | −0.072 ± 0.035 | 0.931 (0.868–0.997) | **4.00E-02** | 3.18E-01 | - | - | - | - | - | - |
| IVW | −0.068 ± 0.036 | 0.934 (0.870–1.003) | 6.20E-02 | 3.18E-01 | 42.401 | 25 | 1.63E-02 | - | - | - |
| MR Egger | −0.092 ± 0.112 | 0.912 (0.732–1.137) | 4.21E-01 | 5.94E-01 | 42.309 | 24 | 1.19E-02 | 0.001 | 0.006 | 8.21E-01 |
| Weighted median | −0.029 ± 0.040 | 0.971 (0.897–1.050) | 4.63E-01 | 6.17E-01 | - | - | - | - | - | - |
| Simple mode | −0.022 ± 0.068 | 0.979 (0.857–1.118) | 7.54E-01 | 8.62E-01 | - | - | - | - | - | - |
| Weighted mode | −0.018 ± 0.058 | 0.982 (0.877–1.100) | 7.53E-01 | 8.62E-01 | - | - | - | - | - | - |
| **Diaphysis** | | | | | | | | | | |
| MR Horse | −0.057 ± 0.035 | 0.945 (0.879–1.012) | 0.103 | 3.18E-01 | - | - | - | - | - | - |
| IVW | −0.127 ± 0.063 | 0.880 (0.779–0.996) | 0.042 | 3.18E-01 | 91.588 | 18 | 7.48E-12 | - | - | - |
| MR Egger | −0.362 ± 0.190 | 0.696 (0.480–1.010) | 0.074 | 3.18E-01 | 83.240 | 17 | 1.01E-10 | 0.015 | 0.012 | 2.09E-01 |
| Weighted median | −0.067 ± 0.041 | 0.935 (0.864–1.013) | 0.098 | 3.18E-01 | - | - | - | - | - | - |
| Simple mode | −0.118 ± 0.064 | 0.889 (0.784–1.007) | 0.081 | 3.18E-01 | - | - | - | - | - | - |
| Weighted mode | −0.092 ± 0.054 | 0.912 (0.820–1.014) | 0.106 | 3.18E-01 | - | - | - | - | - | - |
| **Spine** | | | | | | | | | | |
| MR Horse | 0.034 ± 0.041 | 1.035 (0.953–1.122) | 4.07E-01 | 5.94E-01 | - | - | - | - | - | - |
| IVW | 0.006 ± 0.045 | 1.006 (0.921–1.100) | 8.87E-01 | 9.68E-01 | 105.085 | 34 | 3.48E-09 | - | - | - |
| MR Egger | −0.005 ± 0.179 | 0.995 (0.701–1.413) | 9.78E-01 | 9.78E-01 | 105.071 | 33 | 1.92E-09 | 0.001 | 0.009 | 9.48E-01 |
| Weighted median | 0.063 ± 0.042 | 1.065 (0.980–1.158) | 1.37E-01 | 3.41E-01 | - | - | - | - | - | - |
| Simple mode | 0.038 ± 0.082 | 1.039 (0.885–1.220) | 6.44E-01 | 8.13E-01 | - | - | - | - | - | - |
| Weighted mode | 0.078 ± 0.062 | 1.081 (0.957–1.221) | 2.19E-01 | 4.38E-01 | - | - | - | - | - | - |

Exposure: BMFF (PMID: 39747859); Outcome: T2D (GCST90018926). Number of SNPs for clump $r^2$ < 0.001: femoral head=15; total hip=26; diaphysis=19; spine=35. Heterogeneity was assessed by the inverse-variance weighted (IVW) method and by MR Egger. Pleiotropy was assessed from MR Egger. All tests were two-sided. MR applied the FDR-Q correction (Q value < 0.05 as statistically significant).
P values < 0.05 are shown in bold.

The strength of the associations with T2D is further demonstrated by our sensitivity Obs-PheWAS, in which T2D is the most-significant association for femoral head, total hip and spine, and the second most-significant for diaphysis BMFF. Even when focusing on incident cases, T2D is among the diseases most strongly associated with spine BMFF, with an OR similar to that for osteoporosis. Our MR results further suggest that genetic predisposition to increased BMFF at the spine, total hip, and diaphysis has direct causal associations with T2D that mirror the site-specific PheWAS relationships. Importantly, these causal associations are significant only when using the more lenient LD threshold, again underscoring the need for future, larger-scale BMFF GWAS to allow stronger IVs to be established. Nevertheless, it is

intriguing to speculate how femoral and spinal BMFF might exert divergent effects on the pathogenesis of T2D. One possibility is that femoral BMAT is a site of safer, more metabolically inert lipid storage, analogous to gluteo-femoral adipose tissue, whereas spinal BMAT is akin to more metabolically active visceral adipose tissue, releasing metabolites and endocrine factors that promote insulin resistance[18]. Clearly, it will be important to establish the differences between spinal and femoral BMAT, both on molecular and functional levels, to better understand the relationships between BMAT and T2D.

In addition to osteoporosis and fracture, we reveal further associations between BMFF and other musculoskeletal conditions. One of the most consistent relationships is with 'Other disorders of bone and

cartilage' (PheCODE 733), which is positively associated with BMFF in the spine, femoral head, and total hip. This association is complex to interpret because this PheCODE includes 373 different ICD-10 codes, with diseases as diverse as hypertrophy of bone, osteolysis, skeletal fluorosis, chondromalacia, and physeal arrest. It will be important to determine if the BMFF associations are spread across these conditions or are focused on a subset of diseases within this broad category.

Another common musculoskeletal association is osteoarthritis. Compared to osteoporosis and fracture, the relationship between BMAT and osteoarthritis has received relatively little attention[40]. Our findings are important because they reveal that osteoarthritis is positively associated with spine BMFF but negatively with BMFF of the femoral head or total hip. The latter is consistent with one small human study of osteoarthritic patients, which found that proximal femoral BMFF decreases with increasing disease severity[41]. In contrast, increased bone marrow adiposity was observed in the tibial and femoral epiphyses in a mouse model of osteoarthritis[42]. Thus, alternative mouse models may be needed to study the site-specific BMAT-osteoarthritis relationships that exist in humans. Our PRS-PheWAS also shows that monoarthritis is inversely associated with BMFF in the femoral head, suggesting a robust relationship between femoral head BMFF and different forms of arthritis. Understanding whether BMAT directly impacts arthritis incidence and progression will be an important goal for future research.

The earliest studies of bone marrow adipocytes focused on their relationship with hematopoiesis[2,43]. Therefore, it is notable that our sensitivity Obs-PheWAS identifies associations with several hematological diseases, including anemias and neutropenia. The positive association between these diseases and spine BMFF is consistent with longstanding observations of an inverse relationship between hematopoiesis and BMAT expansion[2,43]; however, these diseases are also negatively associated with femoral head BMFF. This disparity may result from site-dependent differences in bone marrow adipocyte function, as reported for regenerative hematopoiesis in mouse models[2,43]. Beyond these diseases, our PheWASes also identify associations with several hematological cancers, as discussed below.

Our findings are also relevant to the links between marrow adiposity and cancer. Preclinical studies suggest that bone marrow adiposity influences tumor development and progression, including for hematological cancers and bone metastases of solid tumors[2,44,45]. In particular, bone marrow adipocytes secrete factors that promote the migration, viability, and/or survival of multiple myeloma cells, leukemic blasts in acute myeloid leukemia, and metastases from breast or prostate tumors[2,44–47]. Bone marrow adipocytes also promote a quiescent state in acute lymphoblastic leukemia cells, helping them to evade chemotherapy[48]. For some of these cancers, altered bone marrow adiposity has been reported in smaller-scale clinical studies[2,44,45,49,50]; however, our study is notable because it identifies associations across such a large population, and for BMFF at multiple sites.

Our sensitivity Obs-PheWAS findings reveal positive associations between total hip BMFF and prostate cancer, and between spine BMFF and both malignant breast cancer and bone metastases. These results are consistent with the above preclinical findings and with one report of increased vertebral bone marrow adiposity in breast cancer patients[49]. The positive association between chemotherapy and spine BMFF, and between 'effects of radiation' and both spine and total hip BMFF, echoes longstanding reports of increased bone marrow adiposity following cancer treatment[2]. Thus, the positive BMFF-cancer associations observed in our sensitivity Obs-PheWAS could represent confounding effects of cancer therapy and/or reverse causality, with these diseases directly increasing bone marrow adiposity. In contrast to the above cancers, we also reveal a negative association between total hip BMFF and Non-Hodgkin Lymphoma. This extends our previous observation of extremely low femoral BMFF in some people with this disease[13]. Notably, unlike for the spine, total hip BMFF is negatively

associated with malignant breast cancer; hence, it is important to take account of skeletal site when considering BMA-cancer relationships.

In addition to these Obs-PheWAS results, our PRS-PheWAS identifies a positive association between diaphysis BMFF and multiple myeloma. This contrasts with one small study that found a trend for lower axial bone marrow adiposity in multiple myeloma patients[50] but extends observations from mouse models that show increased tibial bone marrow adiposity in early-stage myeloma[47]. Thus, the BMFF-myeloma relationship may vary across skeletal sites. This is important because changes in bone marrow adiposity may identify patients at risk of multiple myeloma progression[51]. In contrast, we find no associations with leukemias, despite reports of decreased bone marrow adiposity in chronic lymphoid, acute lymphoid, and chronic myeloid leukemia[2,44,48]; multiple myeloma is also absent from our Obs-PheWAS results. This may reflect insufficient statistical power caused by the low incidence and prevalence of these conditions among the UKBB population[52]. Despite this limitation, we extend the number of neoplastic conditions that are associated with bone marrow adiposity. Therefore, our findings should inform future efforts to determine if bone marrow adiposity directly influences cancer incidence and/or progression.

Unlike the above disease classes, few studies report associations between bone marrow adiposity and cardiovascular or respiratory diseases. By identifying such associations, our PheWAS results greatly extend the clinical relevance of bone marrow adiposity.

Among these diseases, COPD is notable because its positive association with spine BMFF is stronger than for any other incident disease. In contrast, incident COPD is inversely associated with total hip BMFF. This disparity between spine and femoral BMFF is similar to their relationships with several other diseases, including T2D. Given that T2D itself is associated with COPD[53], we speculate that spine and femoral BMFF may have opposing relationships with COPD because of their divergent causal association with T2D.

Several other cardiometabolic diseases are also associated positively with spine BMFF but negatively with femoral BMFF, particularly when prevalent cases are considered. Here, essential hypertension is especially striking because it is strongly associated with BMFF at all four sites. This relationship may involve the interplay between BMAT and hematopoiesis, combined with the positive causal effect of lymphocyte count on blood pressure[54]; if so, BMFF may be associated with hypertension via effects on lymphopoiesis. Finally, the positive relationship between spine BMFF and incidence of several other cardiovascular conditions suggests a more widespread impact of vertebral BMAT on cardiometabolic dysfunction.

Another notable finding is that there are site and sex differences in BMFF-associated diseases. As discussed above, the BMFF-disease relationships frequently differ depending on skeletal site, with many diseases associating positively with spine BMFF but negatively with BMFF at one or more femoral sites. While this is perhaps most notable for T2D, our sensitivity Obs-PheWAS reveals similar divergent site-specific associations for many other conditions across multiple disease classes. Many of these are interrelated disease clusters, including cardiometabolic diseases and other age-associated chronic diseases. Thus, BMFF distribution may be an important factor in the pathogenesis of these complex multi-morbidities.

These findings highlight the critical importance of analyzing BMFF at multiple anatomical locations and underscore the concept that the pathophysiological functions of BMAT are skeletal site dependent[2,20,21]. Indeed, BMAT has been proposed to exist in two broad subtypes, regulated BMAT (rBMAT) and constitutive BMAT (cBMAT), which differ in anatomical location and in their molecular and functional characteristics[2,20,21]. Thus, cBMAT forms earlier in development, contains densely packed adipocytes, predominates at distal skeletal sites, and is less changeable in response to environmental and pathophysiological variables. In contrast, rBMAT forms later in development,

features adipocytes interspersed with hematopoietic red marrow, predominates in the axial skeleton and more-proximal regions of the limbs, and more readily increases or decreases in response to varying environmental stimuli and pathophysiological states. These characteristics are broadly consistent with our observations, with spine BMFF corresponding to rBMAT and femoral BMFF being more cBMAT-like. However, some disease associations are even stronger for femoral BMFF than for spine BMFF, suggesting that cBMAT is not always more inert than rBMAT. The three femoral sites also differ from one another, suggesting that this binary cBMAT/rBMAT classification is an oversimplification. It will be important for future studies to further dissect these site-specific differences on molecular and functional levels.

Disease risk factors, incidence and outcomes also differ between males and females[55], including for the relationships between body composition and cardiometabolic health[56,57]. Our present findings extend understanding of these phenomena by identifying the extent of sex differences in BMFF-associated diseases. Many associations occur in both sexes, especially for spine BMFF; in such cases, the direction of the BMFF-disease association is the same in each sex. In contrast, for the femoral sites, there are many more associations in females than in males. Sometimes this is because the disease itself is sex-specific, such as those affecting reproductive organs; this is particularly common when prevalent cases are included. However, most of the sexually dimorphic associations among incident cases occur for diseases that can affect males and females. For example, acute renal failure is identified only in males, while the positive association between BMFF and incident osteoporosis occurs for all four sites in females but none in males, even when prevalent cases are included. The latter is reminiscent of previous observations that vertebral bone marrow adiposity is associated with incident bone loss in elderly women, but not in elderly men[26]. Together, our findings highlight sex differences in the clinical impact of bone marrow adiposity.

We acknowledge that our study has several limitations. First, the UKBB MRI data allow fat fraction mapping but do not include spectroscopy, and therefore cannot be used to measure lipid saturation and unsaturation[3]. This is notable because vertebral bone marrow lipid saturation is positively associated with risk of incident fracture in older adults[58] and with prevalent fracture in T2D[27]. The latter, small-scale study found no relationship between fracture and overall bone marrow adiposity, suggesting that bone marrow lipid saturation is an even stronger driver of fracture risk. Bone marrow lipid saturation is also higher in T2D[4]; however, whether this has a direct causal effect on diabetes is unknown.

Secondly, while vertebral fractures are estimated to affect up to 25-50% of people over the age of 50, many are asymptomatic, with only 30-40% coming to medical attention[59,60]. This likely explains why the positive association between BMFF and vertebral fracture is detected only in our sensitivity Obs-PheWAS: while reverse causality may play a role, the number of fracture cases in the incident Obs-PheWAS is too few to detect these associations (Supplementary Data 35). Underreporting of vertebral fractures may also account for our MR results, which show that the causal association of BMFF with fracture is weaker for the spine than for each femoral site.

The challenge of insufficient incident cases also applies to other diseases. The UKBB population is relatively healthy[52], which can limit the ability to detect associations and causal effects of BMFF on many health outcomes. As for vertebral fracture, having insufficient cases may explain why many associations, such as multiple myeloma, are detected in our sensitivity Obs-PheWAS and/or PRS-PheWAS, but not the incident Obs-PheWAS. It may also explain some differences in our sex-stratified Obs-PheWAS. For example, BMFF is associated with osteoporosis in females but not in males; this may result from genuine sex differences in disease pathogenesis but could also reflect males' lower incidence of osteoporosis[23], which would limit the power to detect this association. However, it is important to acknowledge that

reverse causality may contribute to the additional associations identified when prevalent cases are included. We will overcome the challenge of insufficient cases by measuring BMFF in the remaining ~50,000 UKBB imaging participants and by applying our BMFF PRSs to other large-scale genomic cohorts, including meta-analyses of multiple datasets, to further explore the clinical impact of BMAT.

Thirdly, medication use may influence some of the observed BMFF-disease associations. For example, glucocorticoids are used to manage COPD, and they also increase BMFF[2]. However, UKBB participant medication history is mostly from self-reported data, with primary care records being available for only a subset of participants; thus, it is not possible to systematically control for use of medications that may influence the observed associations between BMFF and disease outcomes.

Fourthly, we defined PheCODES based on national medical records (inpatient hospital episode records, cancer registry, and death registry) in UKBB, which did not include diagnoses from primary care. This may result in incomplete case ascertainment for conditions predominantly diagnosed and managed in the primary care setting.

Fifthly, we generated PRSs using PRS-CT (clumping + $p$ value thresholding: to generate weighted PRS based on SNP associated with BMFF) because it is more interpretable for causal inference. However, the inclusion of correlated SNPs reduces this advantage. The PRS-CS method, which utilizes a Bayesian regression framework by placing a continuous shrinkage (CS) prior on SNP effect sizes[61], would be valuable for future work characterizing the full polygenic architecture of BMFF.

There are also some limitations to our MR analysis. MR relies on genetic instruments (SNPs) associated with BMFF, but the proportion of variance ($R^2$) explained by the genetic instruments ranged from 0.06% to 0.41% and the F-statistic ranged from 29.82 to 179.79 across the four bone regions. This limited variance suggests potential weak instrument bias, which could lead to underpowered or biased causal estimates; this is particularly true using the more stringent LD clumping threshold ($r^2 < 0.001$), which found causal associations for total hip or diaphysis BMFF and osteoporosis, but not for any other BMFF-disease associations. The MR analysis assumes that the genetic instruments affect the outcome (osteoporosis, fractures, T2D) only through the exposure (BMFF). However, horizontal pleiotropy, where genetic variants influence the outcome through pathways other than BMFF, cannot be entirely ruled out. Our study used MR-Egger and MR-PRESSO to test for pleiotropy. MR-Egger and MR-PRESSO tests suggested minimal pleiotropy, while the presence of heterogeneity across different bone regions suggested some genetic variants may have pleiotropic effects, potentially biasing the causal estimates. However, a major strength of our MR analysis is that we also used MR-Horse, which is protected from bias due to pleiotropy[62]. Relating to this, a potential limitation of our conservative exclusion of BMD- and T2D-associated variants is that, while this approach reduces the risk of horizontal pleiotropy, it may also lead to a reduction in IVs strength. However, excluding these variants in our MR analyses had minimal impact on overall IVs strength (Supplementary Data 37). Moreover, Steiger filtering[63] confirmed correct causal directions for all variants, including those associated with both exposure and outcome (Supplementary Data 38-41). In addition, although the sample overlap between exposure and outcome GWAS was minimal (< 5%), residual bias cannot be entirely ruled out. We seek to validate our MR findings in the independent dataset once these data are available. A final limitation is that the MR analysis was conducted in a predominantly white European population from the UK Biobank, which may limit the generalizability of the findings to other ethnic groups.

Finally, the UKBB imaging study has plans to include repeat imaging of 60,000 participants, but does not yet have longitudinal MRI data. Such data would be necessary to determine whether changes in BMFF precede disease onset or are a consequence of disease

progression. This will provide critical insights into the temporal dynamics of BMFF alterations and their role in disease incidence and progression.

Despite these limitations, our study is a breakthrough in understanding the fundamental biology and clinical implications of bone marrow adiposity. Our large-scale MR-PheWAS identifies the overall and sex-specific disease outcomes associated with BMFF and its genetic liability at multiple skeletal sites. To explore the short-term effects of BMFF, we conducted Obs-PheWAS, analyzing 1808 disease outcomes using deep-learning-quantified BMFF measurements obtained at the time of MRI scanning. Complementing this, PRS-PheWAS establishes the long-term effects by evaluating the cumulative genetic risk for elevated BMFF levels. Our Obs-PheWAS and PRS-PheWAS not only enhance statistical power and reduce bias but also validate the findings and strengthen causal inference.

The unexpected breadth of BMFF-associated diseases highlights the potential of BMFF as a biomarker for the improved prediction, prevention and treatment of diverse diseases. This includes leveraging our BMFF PRSs as genetic biomarkers, which could benefit clinical practice by eliminating the need for expensive and time-consuming MRI analyses. Our results provide evidence of causality for altered bone marrow adiposity on osteoporosis, fracture, and T2D, which suggests that modulating bone marrow adiposity could be a viable therapeutic strategy for the prevention and treatment of these diseases. Together, our study reveals extensive insights into BMAT function that have the potential to benefit human health.

## Methods

### Ethics

UKBB has ethics approval from the National Health Service North-West Center Research Ethics Committee (Ref11:/NW/0382). All participants provided informed consent. Data for this work were obtained under an approved UKBB project application (ID 48697).

### Study population

UKBB is a large population-based prospective cohort study of 502,352 participants aged 40–60 years recruited between 2006 and 2010[64]. In this study, we used participant data from the ongoing UKBB multi-modal imaging study, initiated in 2014, which comprises brain, cardiac, and abdominal MRI, carotid ultrasound scan, and whole-body DXA[14].

We developed and validated a light-weight attention-based U-Net model for simultaneous detection and segmentation of tiny structures in large 3D MRI imaging data, to generate volumes of interest (VOIs) corresponding to bone marrow regions, including femoral head, total hip, femoral diaphysis, and the thoracic and lumbar spine (Python code [https://doi.org/10.5281/zenodo.13959673]). The segmented VOIs were then applied to the fat fraction (FF) maps. The deep-learning BMFF measurements of the four bone regions were conducted in two batches based on the availability of MRI data released by UKBB ($N = 50,226$). Method details for deep-learning training, development and validation are described in our previous publication[13].

### BMFF polygenic risk score

We conducted meta-GWAS analyses (combining the two batches of BMFF measurements) in the population of unrelated white UKBB participants to identify genetic variants associated with the BMFF of each bone region (method details are described in our previous publication[15]). We generated BMFF polygenic risk scores (PRS) for each bone region using formula 1:

$$PRS_j = \sum_{i}^{N} \beta_i \cdot dosage_{ij}$$

where N is the number of SNPs in the score, $\beta_i$ is the effect size (beta coefficient) of variant $i$ and $dosage_{ij}$ is the number of copies of SNP $i$ in

the genotype of individual $j$. We followed the pipeline in UKBB developed by Collister et al.[65] and used PLINK 2.0[66]. Each SNP effect estimate derived from our BMFF meta-GWAS is referred to as the '$\beta_i$'. To be specific, the PRS for each bone region was constructed by adding up the number of BMFF-increasing alleles for each SNP, weighted for the SNP effect size on BMFF level (beta coefficients from meta-GWAS).

The quality control of the SNPs and samples used to generate PRS for each bone region is summarized in Supplementary Data 11-13. In SNP quality control, we retained significant SNPs (linkage disequilibrium LD $r^2 < 0.6$, P value $< 5 \times 10^{-8}$) found in meta-GWAS, with missingness <0.05, Hardy–Weinberg equilibrium test $P$ value $> 10^{-12}$, non-multiallelic, imputation quality (INFO) > 0.4, and minor allele frequency > 0.005. We further excluded ambiguous AT/CG variants, and insertion/deletion polymorphisms (indels). The PRS were first calculated based on 48 SNPs for femoral head, 115 for total hip, 99 for diaphysis, and 119 for spine (Supplementary Data 14). In addition, we conducted a sensitivity PRS analysis, which further excluded SNPs associated with BMD from the instrumental variants selected. LD proxies ($r^2 > 0.6$) for significant SNPs passing quality control were generated using the LDproxy function in the LDlinkR package (https://github.com/CBIIT/LDlinkR (version 1.4.0) with 1000 Genomes Project Phase 3 as LD reference panel). We searched the GWAS catalog (https://www.ebi.ac.uk/gwas/) to identify the most recently published GWAS with the largest sample size for BMD of each bone region (BMD GWAS details are presented in Supplementary Data 42). A GWAS for diaphysis BMD was not found. The BMD GWASes for the remaining three bone regions (femoral head, total hip, spine) reported distinct significant SNPs associations (Supplementary Data 43). We cross-compared BMFF LD proxies and BMD SNPs and excluded any overlapping SNPs from the list of instrumental variants to construct the PRS. In the PRS sensitivity analysis, the PRS were calculated based on 46 SNPs for femoral head, 114 for total hip, and 117 for spine (Supplementary Data 15). The PRS for diaphysis was not updated due to the absence of a corresponding GWAS or overlaps of instrumental variants in the remaining three bone regions. The variance explained by genetic instrumental variants ($R^2$) for each PRS was estimated following formula 2:

$$R^2 = 2 \times EAF \times (1 - EAF) \times \beta^2$$

Here, EAF represents the effect allele frequency and β denotes the SNP beta coefficient[67].

In sample quality control, we sub-grouped participants based on their ancestry as 'White' using the UKBB data-field 22006 ("Genetic ethnic grouping") and excluded UKBB withdrawals, individuals who were outliers for heterozygosity or missing rates, individuals with a missing rate >0.02 on autosomes, individuals with sex discordance (between the phenotypic and genetically inferred sex), individuals who were not in a maximal set of unrelated individuals up to 3rd degree, and participants included in our meta-GWAS. The study retained 305,313 participants for femoral head, 305,453 for total hip, 306,222 for diaphysis, and 303,066 for spine (Supplementary Data 13).

In summary, we generated two PRSs for each bone region as presented in Supplementary Data 14-15.

**Phenome-wide association analyses.** We performed a PheWAS to identify disease outcomes associated with BMFF in four bone regions. Disease outcomes were defined based on the PheCODE schema (with 10,750 unique ICD-10 codes and 3113 ICD-9 codes) using national medical records (inpatient hospital episode records, cancer registry, and death registry) until 31 March 2023. The PheCODE schema combines correlated ICD codes into a distinct code based on the Phemap v1.2 (https://phewascatalog.org/phecodes_icd10) and excludes patients with related diseases from the corresponding control groups[68]. Herein, we defined cases based on the presence of at least one ICD code mapping to each PheCODE, consistent with our previous

publications[69–73]. We translated the ICD codes into PheCODE groups, which included 1808 PheCODEs classified into 17 disease categories[70,74–76]. Further details of the PheCODE schema and application are provided in the Supplementary Information (Supplementary Note 1).

In Obs-PheWAS, BMFF was measured by our deep-learning models and is presented as rank-transformed (normalized) BMFF for each bone region. The quality control of Obs-PheWAS sample is presented in Supplementary Data 1. We used deep-learning BMFF measurements as the exposure and included only incident cases (defined as disease diagnosed after at least six months of undergoing UKBB MRI imaging) in the PheWAS analysis. We used multivariable logistic regression models to test the associations between rank-transformed BMFF and each PheCODE, with adjustment for age at imaging, sex, and BMI at imaging. We performed the analysis only for those PheCODEs with >200 incident cases, the threshold suggested by the power estimates for PheWAS analysis[77]. We also conducted a sensitivity analysis to include both incident and prevalent cases. We corrected the P values for multiple testing using the false discovery rate (FDR) *q* value threshold of 0.05[78].

In PRS-PheWAS, BMFF PRS were generated for each bone region. We used genetically proxied BMFF level as the exposure and included PheCODEs with >200 incident and prevalent cases. We used multivariable logistic regression models to test the associations between BMFF PRS and each PheCODE, adjusting for age, sex, assessment center, and the first 10 genetic principal components (PC). We applied the FDR correction (*q* value < 0.05 as statistically significant) following the method by Benjamini-Hochberg to account for multiple testing in each of the phenotype-wide analyses[78].

To investigate sex-specific associations in each bone region, we performed sex-stratified Obs-PheWAS and PRS-PheWAS analyses.

The analyses were performed using the 'PheWAS' package [https://github.com/PheWAS/PheWAS] (R version 4.4.1).

**Mendelian randomization analyses**. We conducted two-sample MR analyses to investigate the potential causal associations between BMFF (exposure) and the risk of osteoporosis, fractures and T2D (outcomes). We followed the STROBE-MR (strengthening the reporting of observational studies in epidemiology using Mendelian randomization) guidelines (Please see Supplementary Note 2 in the Supplementary Information file).

We adhered to the three core genetic instrumental variants (IV) assumptions: (1) Relevance: the IV is associated with the exposure of interest; (2) Independence: the IV is independent of any confounders of the exposure-outcome association; and (3) Exclusion Restriction: the IV selected for MR analysis are not associated with the outcome of interest independently of the exposure[79]. To meet the assumptions, we obtained IVs by filtering for BMFF meta-GWAS genome-wide significant SNPs ($P < 5 \times 10^{-8}$), followed by linkage disequilibrium (LD)-based clumping using the 'TwoSampleMR' R package with the following parameters: clump_kb = 10,000, clump_$r^2$ for two thresholds ($r^2 = 0.001$ or $r^2 = 0.6$), pop = "EUR", and excluded those SNPs associated with BMD or T2D to ensure that any observed association between IVs and outcomes is mediated through the exposure (BMFF), maintaining the validity of causal inference. The strength of genetic IVs was tested with the F-statistic (https://shiny.cnsgenomics.com/mRnd/). The F-statistic was calculated using formula 3:

$$F = \frac{\beta^2}{SE^2}$$

Here, $\beta$ represents the beta coefficient of SNP on the exposure, and SE is the standard error of the beta[80]. The F-statistic was estimated to assess whether there was a possibility of weak instrument bias. The genetic instruments for each outcome were derived from summary statistics of the most recent publicly available GWASes of European ancestry with the largest sample size, obtained from the MRC IEU GWAS database (https://gwas.mrcieu.ac.uk/). The identifier for the osteoporosis GWAS[81] is ebi-a-GCST90038656, for the fractures GWAS[81] is ebi-a-GCST90038703, and for the type 2 diabetes GWAS[82] is ebi-a-GCST90018926. BOLT-LMM coefficients, if reported in the outcome GWASes, were transformed to log-odds scale. Summary statistics of the SNP-exposure association and the SNP-outcome association were combined to estimate the causal effect of the exposure on the outcome. SNP effect data for the exposure and outcome were harmonized to ensure consistency in effect direction and allele matching. Harmonization ('harmonise_dat' function in 'TwoSampleMR' package) involved aligning the effect alleles across exposure and outcome datasets, correcting for strand mismatches, and excluding palindromic SNPs with ambiguous allele frequencies.

In MR analyses, six methods were employed to examine the causal association between exposure and outcome, including IVW, MR-Horse[62], MR-Egger, weighted mode, weighted median, and simple model, among which IVW and MR-Horse were the primary methods[83]. MR-Horse is a Bayesian model with horseshoe shrinkage that can account for valid instruments, uncorrelated and correlated pleiotropy[62]. Horizontal pleiotropy tests were performed based on (i) MR-Egger regression (intercept close to 0, P value > 0.05: no evidence for horizontal pleiotropy), and (ii) MR-PRESSO (detected and corrected for horizontal pleiotropy by identifying outlier variants)[84]. The heterogeneity test was estimated based on Cochran's Q statistic. To further assess the robustness of the MR findings, we used the leave-one-out analysis to investigate the influence of outlying and/or pleiotropic genetic variants[85]. MR results were visualized using scatter plots (to display the consistency of causal estimates across different MR methods, and the overall direction of causality), forest plots (to present the causal effect estimates of individual SNPs), funnel plots (to detect asymmetry, which could indicate horizontal pleiotropy or publication bias), and leave-one-out plots (to test IV outliers on the causal estimates).

In addition, we performed sensitivity analyses using IVs derived from the BMI-unadjusted BMFF meta-GWAS (LD clumping threshold: $r^2 < 0.001$) to assess potential collider bias in our primary MR results.

All tests were two-sided. MR applied the FDR-Q correction (*q* value < 0.05 as statistically significant). The analysis was performed using the 'TwoSampleMR', 'MR-PRESSO', and 'R2jags' packages [https://github.com/MRCIEU/TwoSampleMR] (R version 4.4.1).

**Reporting summary**
Further information on research design is available in the Nature Portfolio Reporting Summary linked to this article.

## Data availability
All data for BMFF and BM segmentation volumes have been uploaded to the UKBB (upload ID 5858), where they will be available under UKBB Category 105 to any individuals with an approved UKBB project. Researchers can apply for UKBB access via the UKBB Access Management System (https://ams.ukbiobank.ac.uk/ams/). PheCODEs were generated using data from the Phemap v1.2 (https://phewascatalog.org/phecodes_icd10). PRSs were constructed using data from our previous BMFF meta-GWAS[15] (https://doi.org/10.1038/s41467-024-55422-4). To exclude any SNPs independently associated with BMD or type 2 diabetes, we searched GWAS catalog (https://www.ebi.ac.uk/gwas/) to identify the most recently published GWAS with the largest sample size for BMD or type 2 diabetes; these studies are described in the manuscript. The remaining data are reported in the Supplementary Data files.

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

## Acknowledgements

This work was supported by grants from the Medical Research Council (MR/S010505/1 to W.P.C. and E.T.) and the British Heart Foundation (RE/18/5/34216 for salary support to W.X.; RG/16/10/32375 to support C.W.; 4-year PhD studentship FS/4yPhD/F/22/34175 C for S.S.). W.P.C. was further supported by a Chancellor's Fellowship from the University of Edinburgh. C.G. and T.M. were supported by the Edinburgh Clinical Research Facility and NHS Lothian R&D. E.T. was supported by a Cancer Research UK Career Development Fellowship (C31250/A22804). This work uses data provided by patients and collected by the NHS as part of their care and support. Finally, we are grateful to Dominic Job (Edinburgh Imaging, University of Edinburgh) for support with IT infrastructure, including GPU servers. Rights Retention Statement: For the purpose of open access, the authors have applied a Creative Commons Attribution (CC-BY) license to any Author Accepted Manuscript version arising from this submission.

## Author contributions

Conceptualization, W.P.C.; data curation, W.X., D.M.M., C.W., S.S., G.P., L.W., and W.P.C.; formal analysis, W.X., D.M.M., C.W., S.S., G.P., C.D.G., and W.P.C.; funding acquisition, S.I.K.S., T.M., E.T., and W.P.C.; investigation, W.X., I.M.E., D.M.M., C.W., S.S., G.P., E.T., and W.P.C.; methodology, W.X., I.M.E., D.M.M., C.W., S.S., G.P., C.D.G., S.B., J.P., L.W., X.L., P.R.H.J.T., M.T., S.I.K.S., T.M., E.T., and W.P.C.; project administration, S.I.K.S., T.M., E.T., and W.P.C.; resources, S.I.K.S., T.M., E.T., and W.P.C.; software, D.M.M., C.W., G.P.; supervision, S.I.K.S., T.M., E.T., and W.P.C.; visualization, W.X. and W.P.C.; writing—original draft, W.X., S.S., E.T. and W.P.C.; writing—review & editing, W.X., I.M.E., G.P., J.P., M.T., S.I.K.S., E.T., and W.P.C.

## Competing interests

G.P. is currently an employee of Pfizer; however, Pfizer had no role in the design or interpretation of this research. All other authors declare no competing interests.

## Additional information

¹Centre for Global Health, Usher Institute, University of Edinburgh, Edinburgh, UK. ²Institute for Neuroscience and Cardiovascular Research, The University of Edinburgh, Edinburgh, UK. ³Edinburgh Imaging, University of Edinburgh, The Queen's Medical Research Institute, Edinburgh BioQuarter, 47 Little France Crescent, Edinburgh, UK. ⁴School of Mathematics and Computer Sciences, Heriot-Watt University, Edinburgh, UK. ⁵Archimedes Unit, Athena Research Centre, Artemidos 1, Marousi, Greece. ⁶Univ. Lille, CHU Lille, Marrow Adiposity and Bone Laboratory (MABlab) ULR 4490, Department of Rheumatology, Lille, France. ⁷Department of Big Data in Health Science, School of Public Health and The Second Affiliated Hospital, Zhejiang University School of Medicine, Hangzhou, China. ⁸Medical Research Council Human Genetics Unit, Medical Research Council Institute of Genetics & Molecular Medicine, University of Edinburgh, Edinburgh, UK. ⁹Danish Institute for Advanced Study (DIAS), Epidemiology, Biostatistics and Biodemography, Department of Public Health, University of Southern Denmark, Odense, Denmark. ¹⁰Centre for Clinical Brain Sciences, University of Edinburgh, The Queen's Medical Research Institute, Edinburgh BioQuarter, 47 Little France Crescent, Edinburgh, UK. ¹¹Edinburgh Cancer Research Centre, Institute of Genetics and Cancer, University of Edinburgh, Edinburgh, UK. ✉e-mail: E.Theodoratou@ed.ac.uk; W.Cawthorn@ed.ac.uk

