## [Transparent Peer Review file · Nature Communications]

Clinical implications of bone marrow adiposity identified by phenome-wide association and Mendelian randomization in the UK Biobank

Corresponding Author: Dr William Cawthorn

Version 0:

Reviewer comments:

Reviewer #1

(Remarks to the Author)

In this manuscript, Xu and colleagues, in a follow up of their previous study studying the genetics underlying BMFF, now attempt to link variations in BMFF to different diseases, by performing a PheWAS in UKBB. This is an under-studied adipose depot, and this paper will therefore be a valuable resource to the field.

I have just a few questions, probably emerging from my own ignorance. However, some of these may well be useful to address either in the introduction or the discussion.

1. The authors show that different BMFF depots confer risk/protection from different diseases, mirroring what we see for different adipose tissue depots. What is known about the bone marrow of different bones?
2. The BMFF from the vertebrae consistently are associated with many more diseases, why might this be the case? Is there a bona fide biological explanation for it, are there potential technical confounders in visualizing the BMFF in these bones?
3. Do the authors have an opinion about the strong link of all four BMFF depots with osteoporosis? Is it something that is being secreted?

Reviewer #2

(Remarks to the Author)

The authors present their investigation into diseases associated with bone marrow fat fraction (BMFF) at four anatomical sites (spine, femoral head, total hip, and femoral diaphysis) using UK Biobank data. For this, they build on their previously developed deep learning algorithm to quantify BMFF from UK Biobank MRI scans and corresponding discovery GWAS (refs 13/15). The current manuscript expands on those findings by (i) examining the association between measured BMFF and up to 1,808 phecodes ("Obs-PheWAS"), and using their GWAS summary statistics to (ii) generate weighted scores of top hits to conduct additional PheWAS ("PRS-PheWAS") and (iii) performing two-sample Mendelian randomization (MR) analyses on three outcomes (osteoporosis, fractures, and type 2 diabetes). The PheWAS are performed separately in a sex-combined and sex-stratified manner and for both incident and prevalent disease. In addition, for (ii), a second set of scores are created for sensitivity analyses by excluding variants which have been previously found to associate with body mineral density (BMD). Overall, the manuscript reports a substantial volume of novel results – sometimes at the expense of readability – but generally provides sufficient documentation, as evidenced by the extensive supplementary materials and comprehensive discussion.

However, I have serious reservations regarding the MR analyses and their interpretation:

- the GWAS results for osteoporosis and fractures appear to be from Dönertaş et al. (PMID 33959723), which used BOLT-

LMM. Due to BOLT-LMM modelling binary traits as continuous outcomes, its effect sizes require transformation for appropriate use in MR. This transformation substantially impacts the scaling of the estimates – could the authors confirm they did not overlook this?

- the authors' conclusions appear to hinge largely on nominal significance of p-values ($p < 0.05$), without adjusting for multiple testing across exposures, outcomes, and MR estimators. More critically, the lack of clinical interpretation of effect sizes can lead to misleading causal claims when essentially null or negligible effects are detected with high precision (i.e., most of Figures 4 and 5). [though, as mentioned, these effects might reflect scaling issues]
- MR methods generally assume the use of independent instruments. However, the authors include SNPs with a LD $R^2 < 0.6$, far above standard thresholds (e.g., 0.01 or even 0.001, the latter typically requiring a large LD reference panel). While methods exist to account for correlated variants (e.g., via LD matrices), it appears these were not applied, which may lead to overly narrow confidence intervals.

Several other questions and suggestions:

- did the PRS-PheWAS exclude individuals who were part of the BMFF-GWAS? If not, could this lead to overfitting or bias?
- please clarify whether FDR correction in the PheWAS analyses was applied within each analysis set or also across stratifications (e.g., sex, incident vs. prevalent disease, PRS1 vs. PRS2).
- what was the rationale for including BMI included as a covariate in the Obs-PheWAS?
- many results are duplicated in both tables and figures. Please consider reducing redundancy.
- please detail the PheCode generation process. Were default settings used (e.g., ≥ 2 ICD codes to define a case, individuals with only 1 code excluded)? Were male- and female-specific phecodes generated, and if so, how were they handled?
- how much sample overlap existed between exposure and outcome datasets in the MR analyses? Could this introduce bias?
- please check whether the GWAS on osteoporosis and fractures included EUR ancestry participants only.
- while simpler PRS are more interpretable for causal inference, the inclusion of correlated SNPs reduces this advantage. If the goal was instead to characterise the broader phenotypic profile of innate BMFF predisposition through the PRS-PheWAS, wouldn't more advanced methods (e.g., PRS-CS, LDpred, SBayesRC) be more suitable?

Minor

- lines #236-237: please include the scaling used for BMFF in its GWAS when interpreting the effect sizes. The reported coefficients (e.g., 0.189 to 0.381) are otherwise uninterpretable. Also, are these log-odds?
- what LD reference panel was used when assessing LD R^2 among SNPs?
- the term polygenic risk score (PRS) is technically correct but now often connotes genome-wide, highly polygenic approaches. Since your score is based on genome-wide significant hits, please clarify this in the abstract and main text.
- a few suggestions to improve readability of your tables: (i) remove log-odds if you already present ORs and 95% CIs, (ii) reduce decimals to 1-2 to avoid pseudo-precision, (iii) format sample sizes with commas (e.g., 10,000)
- from your GWAS publication, it appears you ran both BMI-adjusted and unadjusted GWAS. Which version was used for score creation and MR? If the BMI-adjusted version was used, could collider bias influence your results?
- for datasets obtained via MRC IEU, please consider including references to the original GWAS studies.
- please check STable 17 for remaining notes that should be removed.

Reviewer #3

(Remarks to the Author)

This manuscript is a comprehensive analysis investigating diseases associated with bone marrow adipose tissue (BMAT). This analysis builds on previously published work that used deep learning to measure BMFF of 4 sites (femoral head, total hip, femoral diaphysis, spine) of >44,000 participants in the UK Biobank and genome-wide association studies. The authors now examine BMAT associations using phenome-wide association studies (PheWAS) through different methods: an Observational-PheWAS, a polygenic risk score (PRS)-PheWAS, and Mendelian Randomization.

In the Observational-PheWAS, BMFF was associated with 47 incident diseases across 12 disease categories, which included osteoporosis, fracture, type 2 diabetes, and also diseases not previously linked to BMAT, such as cardiovascular diseases, several cancers, and other conditions. The PRS-PheWAS and Mendelian Randomization analyses provide additional evidence and causality for a relationship between BMAT and osteoporosis, fractures, and T2D.

There is growing appreciation for the importance of BMAT, a unique and dynamic fat depot, that has significant implications on skeletal and systemic metabolism and nutrition. This manuscript addresses a significant gap in the comprehensive and large-scale evaluation of the physiologic and pathologic roles of BMAT. Significant strengths include methods using multiple approaches to evaluate the relationship of BMAT to PheWAS (Observational-PheWAS and PRS-PheWAS), Mendelian Randomization to infer causality, and a thoughtful approach to specifically evaluate osteoporosis diagnoses, fracture, and findings independent of BMD. Another significant strength is the large sample size using a gold standard method to analyze BMAT and analyzing 4 skeletal sites, which is valuable given site-specific variations in BMAT. This is a well-written manuscript by leaders in the field. My concerns are noted below.

1. Confounding: The Obs-PheWAS analyses were adjusted for age, sex, and BMI at imaging. The PRS-PheWAS analyses were adjusted for age, sex, assessment centre, and the first 10 genetic principal components. There could be confounding variables that may be influencing associations. For example, glucocorticoid use increases BMAT and is associated with COPD. It may not be possible to account for or adjust for all the various factors, but confounding factors should be considered, especially for BMAT associations with unexpected diseases.

2. Phenome: Given the importance of ICD codes for mapping the phenome, it would be important to provide additional information and note important limitations. The authors note that disease outcomes were defined based on national medical records (inpatient hospital episode records, cancer registry, and death registry). Would this miss ambulatory diagnoses? This may affect the case detection for osteoporosis and diabetes, and other diseases managed in the ambulatory setting.

3. Site-specific BMAT: Given the site-specific nature of these findings, it would be helpful if the authors could briefly elaborate on the biological significance and how these findings could inform our understanding of regulated and constitutive BMAT.

4. Spine BMAT and skeletal outcomes: There was not an observed causal association for spine BMFF and osteoporosis. Prior studies found that spine BMFF had a robust inverse association with spine BMD. Based on the earlier concerns, if there was an under-detection of osteoporosis at the spine and many vertebral compression fractures are asymptomatic, this may affect the ability to detect this relationship. The authors nicely summarize these limitations later in the discussion, but it may be more helpful to include these earlier when discussing the interpretation of this finding.

Version 1:

Reviewer comments:

Reviewer #2

(Remarks to the Author)

I thank the authors for their detailed replies. From my reading, they have complied with the SAGER guidelines. My remaining, relatively minor comments:

(1) while FUMA uses an initial $r^2 < 0.6$ threshold to define independent signals from GWAS, this is a relatively lenient threshold that results in sets of correlated variants. As the authors rightly note, using such a threshold for instrument selection in Mendelian randomization (MR) leads to spuriously precise estimates and inflated type I error rates. I therefore appreciate the inclusion of results based on more conventional LD thresholds ($r^2 < 0.01$ would also have been appropriate, particularly given the limited resolution of the 1000 Genomes reference panel). However, the rationale for including results based on $r^2 < 0.6$ remains unclear, as the claim that these analyses serve to “demonstrate the impact of IV strength on MR results” seems misleading, as the LD threshold used to define independence does not affect the strength of individual instruments. I suggest moving this sensitivity analysis to the supplementary materials.

(2) the exclusion of variants associated with BMD or T2D may have inadvertently removed some of the strongest instruments for MR, and such exclusions do not “ensure that any observed association between IVs and outcomes is mediated through the exposure.” Approaches such as Steiger filtering would be more appropriate for assessing the direction of causality in cases where variants are associated with both exposure and outcome. The authors may wish to consider this in their future work.

(3) please clarify the reference to PhecodeX, as this represents a different phecode map than v1.2. Additionally, the method by which phecodes were derived remains somewhat unclear. The recommended practice is to define cases as individuals with two or more ICD codes mapping to the same phecode: either two identical codes or two distinct codes that map to the same phecode. Individuals with only one relevant ICD code could then be coded as missing (NA) for that phecode. Despite a previous request for clarification, it remains uncertain whether this approach was followed. Moreover, did the authors use individual-level records (e.g., HES) which capture all ICD code occurrences, or relied on summary diagnosis fields (e.g., UKB Field 41270), which only include distinct diagnosis codes, thereby missing repeated instances of the same ICD code? I do not suggest reanalysing the data if summary fields were used, but it is essential that the methods are clearly described their methods to ensure transparency and reproducibility, and so potential limitations in the current approach can be acknowledged.

Reviewer #3

(Remarks to the Author)

The authors have thoroughly addressed my concerns in the response.

Version 2:

Reviewer comments:

Reviewer #2

(Remarks to the Author)

I thank the authors for their detailed and nuanced responses, which fully addressed the raised comments.

Response to reviewers' comments for NCOMMS-25-16128A: *Clinical implications of bone marrow adiposity identified using deep learning, phenome-wide association, and Mendelian randomization in the UK Biobank*

Reviewer #1 (Remarks to the Author):

In this manuscript, Xu and colleagues, in a follow up of their previous study studying the genetics underlying BMFF, now attempt to link variations in BMFF to different diseases, by performing a PheWAS in UKBB. This is an under-studied adipose depot, and this paper will therefore be a valuable resource to the field.

I have just a few questions, probably emerging from my own ignorance. However, some of these may well be useful to address either in the introduction or the discussion.

1. The authors show that different BMFF depots confer risk/protection from different diseases, mirroring what we see for different adipose tissue depots. What is known about the bone marrow of different bones?

Response:

Thank you for your review and valuable feedback on our manuscript.

Regarding the site-specific differences, an established concept in the study of bone marrow adipose tissue (BMAT) is that there are site-specific BMAT subtypes that have distinct differences in their molecular and functional characteristics. This was first reported in the 1970s and, in 2015, led to the proposal that BMAT exists in two broad subtypes: regulated BMAT (rBMAT) and constitutive BMAT (cBMAT) (Scheller et al. 2015, Craft et al. 2018). cBMAT is composed of more-densely packed adipocytes; forms earlier in development, beginning at distal skeletal sites (i.e. the ends of the limbs); predominates in the appendicular skeleton; and is typically less changeable in response to environmental and pathological stimuli. In contrast, rBMAT comprises clusters of adipocytes interspersed with the hematopoietic red marrow; tends to accumulate later in life; predominates in the axial skeleton (and more proximal regions of the limbs); and varies more in response to pathophysiological variables. Indeed, this may be why we find more variability in spine BMFF (which is more rBMAT-like) than femoral BMFF (which is more cBMAT-like).

While the designation of rBMAT vs cBMAT is conceptually useful, it is now recognised that it is likely an over-simplification and that there are further molecular and functional differences within these categories. Our UK Biobank data further support this, i.e. we find that the genetic architecture of BMFF differs between the three femoral sites (Xu et al. 2025). Therefore, in

our manuscript we initially avoided using the terms rBMAT and cBMAT and instead discussed the concept of site-specific differences, particularly between axial and appendicular bones (i.e. lines 72-74, 109-110, 412-414 and 590-600 of our original submission). However, we have now updated the Discussion with further details about rBMAT and cBMAT (lines 634-648):

“Indeed, BMAT has been proposed to exist in two broad subtypes, regulated BMAT (rBMAT) and constitutive BMAT (cBMAT), which differ in anatomical location and in their molecular and functional characteristics^{2,20,21}. Thus, cBMAT forms earlier in development, contains densely packed adipocytes, predominates at distal skeletal sites, and is less changeable in response to environmental and pathophysiological variables. In contrast, rBMAT forms later in development, features adipocytes interspersed with hematopoietic red marrow, predominates in the axial skeleton and more-proximal regions of the limbs, and more readily increases or decreases in response to varying environmental stimuli and pathophysiological states. These characteristics are broadly consistent with our observations, with spine BMFF corresponding to rBMAT and femoral BMFF being more cBMAT-like. However, some disease associations are even stronger for femoral BMFF than for spine BMFF, suggesting that cBMAT is not always more inert than rBMAT. The three femoral sites also differ to one another, suggesting that this binary cBMAT/rBMAT classification is almost certainly an oversimplification. It will be important for future studies to further dissect these site-specific differences on molecular and functional levels.”

We hope this addresses the reviewer’s question and provides sufficient detail for readers to consider the implications of our findings for increasing understanding of the site-specific nature of BMAT.

2. The BMFF from the vertebrae consistently are associated with many more diseases, why might this be the case? Is there a bona fide biological explanation for it, are there potential technical confounders in visualizing the BMFF in these bones?

Response:

Thank you for this helpful comment. We agree that it is interesting that spine BMFF shows many more disease associations than BMFF at each femoral site. The reasons for this are unclear but are unlikely to relate to technical issues in visualising femoral BMFF (the ability of MRI to detect BMFF does not differ between spine and femur (Karampinos et al. 2018). Instead, this finding is consistent with the above-mentioned concept of rBMAT and cBMAT: BMAT in axial bones is of a ‘regulated’ subtype that changes in different pathophysiological contexts, whereas BMAT in the appendicular skeleton (limbs) is more ‘constitutive’, being more resistant to accumulation or depletion in response to altered environmental or pathophysiological conditions.

Consistent with this, we see that there is much more variability in spine BMFF than in BMFF at the femoral sites. Therefore, it may be that we are able to detect more pathological associations with spine BMFF, in part, because it provides more variation as an exposure.

3. Do the authors have an opinion about the strong link of all four BMFF depots with osteoporosis? Is it something that is being secreted?

Response:

The relationship between BMAT and osteoporosis has been a longstanding topic of interest in the BMAT field: increased BMAT in osteoporosis was first reported in 1971 (Meunier et al. 1971) and there has since been much research into whether BM adipocytes (BMAds) directly influence bone remodelling and pathological bone loss. As you suggest, several BMAd-secreted factors may play a role (reviewed in (Sulston et al. 2016)). For example, BMAds can impair osteoblast function by releasing palmitate or extracellular vesicles that contain adipogenic RNAs. Production of osteoclast-regulating molecules as also been reported, including CXCL1, CXCL2 and RANKL, which stimulate bone resorption. Adipogenic precursors within the bone marrow can also release factors that alter bone remodelling, including DPP4 (Ambrosi et al. 2017) and RANKL (Yu et al. 2020).

In our original submission (lines 409-411) we briefly mentioned this ability of BMAd-secreted factors to regulate bone formation and resorption (“bone marrow adipocytes secrete a range of anti-osteogenic and/or pro-resorptive factors”).

A separate theory relates to BMAds and osteoblasts each being derived from skeletal stem cells, suggesting that any increase in BM adipogenesis will necessarily cause decreased osteoblastogenesis (Gimble et al. 1996). Therefore, osteoporosis may reflect an imbalance in skeletal stem cell fate (something that has been observed in mouse models of ageing, i.e. (Ambrosi et al. 2017)). One very recent study confirms that there are also site-specific differences in human skeletal stem cells (Ambrosi et al. 2025). Therefore, we have updated our revised manuscript to explain that differences in skeletal stem cells may also contribute to the site-specific relationships between BMFF and osteoporosis (lines 434-439):

“Another theory is that, because BM adipocytes and osteoblasts derive from a common progenitor (the skeletal stem cell), any increase in adipogenesis will necessarily come at the expense of osteogenesis². Indeed, site-specific differences in human skeletal stem cells have recently been identified²². Thus, another possibility is that the balance between adipogenesis and osteoblastogenesis varies across skeletal sites, contributing to site-specific differences in the BMFF-osteoporosis relationship.”

Reviewer #2 (Remarks to the Author):

The authors present their investigation into diseases associated with bone marrow fat fraction (BMFF) at four anatomical sites (spine, femoral head, total hip, and femoral diaphysis) using UK Biobank data. For this, they build on their previously developed deep learning algorithm to quantify BMFF from UK Biobank MRI scans and corresponding discovery GWAS (refs 13/15). The current manuscript expands on those findings by (i) examining the association between measured BMFF and up to 1,808 phecodes ("Obs-PheWAS"), and using their GWAS summary statistics to (ii) generate weighted scores of top hits to conduct additional PheWAS ("PRS-PheWAS") and (iii) performing two-sample Mendelian randomization (MR) analyses on three outcomes (osteoporosis, fractures, and type 2 diabetes). The PheWAS are performed separately in a sex-combined and sex-stratified manner and for both incident and prevalent disease. In addition, for (ii), a second set of scores are created for sensitivity analyses by excluding variants which have been previously found to associate with body mineral density (BMD). Overall, the manuscript reports a substantial volume of novel results – sometimes at the expense of readability – but generally provides sufficient documentation, as evidenced by the extensive supplementary materials and comprehensive discussion.

Response:

We thank the reviewer for their thoughtful comments. We are pleased to hear that you recognize the novelty and significance of our MR-PheWAS study of BMFF in UK Biobank.

However, I have serious reservations regarding the MR analyses and their interpretation:

- MR comment 1: the GWAS results for osteoporosis and fractures appear to be from Dönertaş et al. (PMID 33959723), which used BOLT-LMM. Due to BOLT-LMM modelling binary traits as continuous outcomes, its effect sizes require transformation for appropriate use in MR. This transformation substantially impacts the scaling of the estimates – could the authors confirm they did not overlook this?

- MR comment 2: the authors' conclusions appear to hinge largely on nominal significance of p-values ($p < 0.05$), without adjusting for multiple testing across exposures, outcomes, and MR estimators. More critically, the lack of clinical interpretation of effect sizes can lead to misleading causal claims when essentially null or negligible effects are detected with high precision (i.e., most of Figures 4 and 5). [though, as mentioned, these effects might reflect scaling issues]

- MR comment 3: MR methods generally assume the use of independent instruments. However, the authors include SNPs with a LD $R^2 < 0.6$, far above standard thresholds (e.g., 0.01 or even 0.001, the latter typically requiring a large LD reference panel). While methods

exist to account for correlated variants (e.g., via LD matrices), it appears these were not applied, which may lead to overly narrow confidence intervals.

Response:

We appreciate the reviewer's insightful and thorough comments on the MR analysis. Below, we provide a detailed response to each comment, outlining the revisions made to the manuscript. The corresponding updates to the manuscript, tables, and figures are described below under the **'MR revision'**.

MR comment 1: BOLT-LMM Effect Size Transformation for Binary Traits

We performed MR analysis to investigate BMFF as exposure, and Osteoporosis/ Fractures/ Type 2 diabetes as outcomes. The genetic instruments for each outcome were derived from the summary statistics of the most recent publicly available GWASes of European ancestry with the largest sample size, obtained from the MRC IEU GWAS database (<https://gwas.mrcieu.ac.uk/>; <https://www.ebi.ac.uk/gwas/>).

We acknowledge the importance of transforming effect sizes when using BOLT-LMM for binary traits in MR analyses, since MR relies on the assumption that genetic variants are associated with the exposure on a scale compatible with the outcome, particularly for binary traits, where log-odds or liability scales are typically required.

We have therefore transformed the BOLT-LMM coefficients of Osteoporosis- 'GCST90038656'/ Fractures- 'GCST90038703' to log-odds scale.

MR comment 2: Multiple Testing Correction

Thank you for this helpful comment.

We followed previously published MR studies (Sproviero et al. 2021, Liang et al. 2023, Li et al. 2024) to use $p\text{-value} < 0.05$. However, we acknowledge the importance of adjusting for multiple testing to control the false discovery rate (FDR). In response, we have applied the FDR-Q correction ($q\text{-value} < 0.05$ as statistically significant) in MR.

MR comment 3: LD Threshold for Instrument Selection

We thank the reviewer for this thoughtful comment. In our meta-GWAS paper, we used FUMA to identify genome-wide significant SNPs ($P\text{-value} < 5 \times 10^{-8}$) in low LD ($r^2 < 0.6$) as independent significant SNPs. The $P\text{-value}$ and r^2 thresholds we used there are those suggested by FUMA (Watanabe et al. 2017) and used in many previously published

GWASes (Nagel et al. 2018, Jansen et al. 2019, Jansen et al. 2020, Krohn et al. 2022, Salih et al. 2023, Singh et al. 2023).

However, we understand the concern that, for IVs in MR, $r^2 < 0.6$ is lenient, and without stricter corrections for LD, MR analysis may suffer from inflated type I error rates due to overly narrow confidence intervals.

To address this, we have now used LD-clumping with $r^2 < 0.001$ to select a set of independent instruments for each trait. The MR results for $r^2 < 0.001$ and $r^2 < 0.6$ are presented in In-text Tables 5-7: the results with $r^2 < 0.001$ are now considered as the primary analysis, while the results with $r^2 < 0.6$ serve as a sensitivity analysis. We feel it is important and informative to present both r^2 thresholds to demonstrate the impact on the IV strength on MR results.

MR revision:

To address MR comments 1-3, we have made the following revisions:

- **Methods in the manuscript (lines 883-894):**

“We adhered to the three core genetic instrumental variants (IV) assumptions: (1) Relevance: the IV is associated with the exposure of interest; (2) Independence: the IV is independent of any confounders of the exposure-outcome association; and (3) Exclusion Restriction: the IV selected for MR analysis are not associated with the outcome of interest independently of the exposure⁷⁴. To meet the assumptions, we obtained IVs by filtering for BMFF meta-GWAS genome-wide significant SNPs ($P < 5 \times 10^{-8}$), followed by linkage disequilibrium (LD)-based clumping using the ‘TwoSampleMR’ R package with the following parameters: `clump_kb = 10,000`, `clump_r2` for two thresholds ($r^2 = 0.001$ or $r^2 = 0.6$), `pop = “EUR”`, and excluded those SNPs associated with BMD or T2D to ensure that any observed association between IVs and outcomes is mediated through the exposure (BMFF), maintaining the validity of causal inference. The strength of genetic IVs was tested with the F-statistic (<https://shiny.cnsgenomics.com/mRnd/>).”

- **Methods in the manuscript (lines 900-907):**

“The genetic instruments for each outcome were derived from summary statistics of the most recent publicly available GWASes of European ancestry with the largest sample size, obtained from the MRC IEU GWAS database (<https://qwas.mrcieu.ac.uk/>). The identifier for the osteoporosis GWAS⁷⁶ is ebi-a-GCST90038656, for the fractures GWAS⁷⁶ is ebi-a-GCST90038703, and for the type 2 diabetes GWAS⁷⁷ is ebi-a-GCST90018926. BOLT-LMM coefficients, if reported in the outcome GWASes, were transformed to log-odds scale. Summary statistics of the SNP-exposure association and the SNP-outcome association were combined to estimate the causal effect of the exposure on the outcome.”

- Methods in the manuscript (lines 934-935) and in Supplementary Note 1 (point 9):
“All tests were two-sided. MR applied the FDR-Q correction (q-value <0.05 as statistically significant).”
- Results in the manuscript (lines 325-363):
“The causal association of BMFF genetic predisposition with osteoporosis is summarized in Table 5. When using the more stringent LD clumping threshold ($r^2 < 0.001$), positive associations with osteoporosis were found for total hip BMFF and diaphysis BMFF based on the inverse-variance weighted (IVW) method and for diaphysis BMFF based on MR-Horse. Moreover, consistent positive associations were detected for all three femoral sites when the less-stringent LD clumping threshold of $r^2 < 0.6$ was applied. However, no causal association was found in the spine, regardless of the r^2 threshold used.”
“The causal association of BMFF genetic liability with fractures is summarized in Table 6. Each of the MR methods found no significant causal associations for any of the four bone regions when using the more stringent LD clumping threshold ($r^2 < 0.001$). However, when the more lenient LD clumping threshold of $r^2 < 0.6$ was applied, IVW and MR-Horse showed positive associations with fractures for all three femoral sites.”
“The causal association of BMFF genetic liability with T2D is summarized in Table 7. When using the more stringent LD clumping threshold ($r^2 < 0.001$), none of the MR methods found significant causal associations in any of the four bone regions. At the less stringent LD clumping threshold ($r^2 < 0.6$), the findings suggested negative associations with T2D in the total hip and diaphysis, whereas MR-Horse observed a positive association between BMFF genetic liability and T2D in the spine.”
- Figure 1 has also been updated to explain that evidence for causal associations was found primarily for osteoporosis, with weaker evidence for fractures and T2D.
- Discussion in the manuscript (lines 392-395):
“Finally, our MR studies reveal that genetic predisposition to increased BMFF at the total hip and femoral diaphysis is causally associated with osteoporosis and suggest that spine and femoral BMFF may also be causally associated with fractures and T2D.”
- Discussion in the manuscript (lines 723-725):

“...using the more stringent LD clumping threshold ($r^2 < 0.001$)... found causal associations for total hip or diaphysis BMFF and osteoporosis, but not for any other BMFF-disease associations.”

Several other questions and suggestions:

- did the PRS-PheWAS exclude individuals who were part of the BMFF-GWAS? If not, could this lead to overfitting or bias?

Response:

Thank you for this comment. We excluded individuals who were part of the BMFF-GWAS in our PRS-PheWAS. Please find details in Supplementary Table 13 ('Exclude EIDs in meta-GWAS sample'):

	Femoral head	Total hip	Diaphysis	Spine
Sample-QC	337275	337275	337275	337275
Exclude EIDs in meta-GWAS sample	38581	38394	37513	41204
PRS-sample	305313	305453	306222	303066

We explained this further in the Methods section (“BMFF polygenic risk score”, Lines 757-764 of original manuscript; Lines 828-835 of revised manuscript): *“In sample quality control, we sub-grouped participants based on their ancestry as ‘White’ using the UKBB data-field 22006 (“Genetic ethnic grouping”) and excluded UKBB withdrawals, individuals who were outliers for heterozygosity or missing rates, individuals with a missing rate > 0.02 on autosomes, individuals with sex discordance (between the phenotypic and genetically inferred sex), individuals who were not in a maximal set of unrelated individuals up to 3rd degree, and participants included in our meta-GWAS. The study retained 305,313 participants for femoral head, 305,453 for total hip, 306,222 for diaphysis, and 303,066 for spine (Supplementary Table 13).”*

Given the amount of information in our manuscript, we appreciate that this detail may have been easy to miss. We have maintained the above text in our revised manuscript so that readers can interpret our PheWAS analyses.

- please clarify whether FDR correction in the PheWAS analyses was applied within each analysis set or also across stratifications (e.g., sex, incident vs. prevalent disease, PRS1 vs. PRS2).

Response:

We thank the reviewer for this thoughtful comment. Consistent with previously published PheWAS studies (Francis et al. 2022, Korologou-Linden et al. 2022, Khan et al. 2023, Gorman et al. 2024, Rajesh et al. 2025), we applied FDR correction in each set of the PheWAS analyses, rather than across stratifications. The reason for this is that our study aimed to identify disease outcomes associated with BMFF and/or PRS in sex-combined and/or sex-specific UKBB samples. Since PheWAS analysis was conducted separately for each stratification, FDR correction was applied within each PheWAS analysis set to ensure that the correction accounts for the number of tests within each PheWAS, maintaining control over false positives and preserving statistical power. Applying FDR correction across different stratifications (such as male vs. female) would involve treating all tests across the stratifications as a single set of hypotheses, which may not be appropriate if the tests are not independent. Combining all tests across stratifications risks over-correction and dilution of true signals in each stratum. Applying FDR separately ensures that associations in one group are not penalized by non-significant results in another.

- what was the rationale for including BMI included as a covariate in the Obs-PheWAS?

Response:

The primary rationale for including BMI as a covariate in Obs-PheWAS with BMFF is to control for confounding by BMI and obesity. Indeed, BMI is a well-established predictor of numerous disease outcomes (e.g., metabolic, cardiovascular, and musculoskeletal conditions), and our previous study confirms correlations between BMI and BMFF at each site (Xu et al. 2025). Therefore, failing to adjust for BMI could lead to biased estimates, where observed associations between BMFF and disease outcomes (PheCODEs) might be driven by BMI, rather than BMFF itself. This is particularly important in PheWAS, which examines a wide range of outcomes, as BMI's broad impact could mask or exaggerate specific effects of BMFF. By adjusting for BMI, we aim to isolate the independent effect of BMFF on disease outcomes, reducing bias from shared pathways linked to BMI. If BMFF is associated with osteoporosis after BMI adjustment, it would suggest a bone-specific mechanism beyond general obesity.

- many results are duplicated in both tables and figures. Please consider reducing redundancy.

Response:

Thank you for this suggestion. We originally presented several results in both graphical and tabular format because we find that readers often have different preferences in how results are shown. However, we agree that the MR results are best presented as tables and that the

previous figures of these results did not add much beyond the tables. Therefore, in-text figures 4-6 (MR: osteoporosis; fractures; type 2 diabetes) have now been removed.

- please detail the PheCode generation process. Were default settings used (e.g., ≥ 2 ICD codes to define a case, individuals with only 1 code excluded)? Were male- and female-specific phecodes generated, and if so, how were they handled?

Response:

Thank you for this comment. We have added details of the PheCODE generation process in the Supplementary Information file (please see new section entitled “Supplementary Note 2”) and cited references of our previously published PheWAS for PheCODE schema information in the Methods. The relevant text is as follows:

Supplementary note 2:

“Disease outcomes were defined based on the PheCODE schema (with 10,750 unique ICD-10 codes and 3,113 ICD-9 codes) using national medical records (inpatient hospital episode records, cancer registry, and death registry) until March 31, 2023. The PheCODE schema combines correlated ICD codes into a distinct code based on the Phemap v1.2 (https://phewascatalog.org/phecodes_icd10) and excludes patients with related diseases from the corresponding control groups. To account for the correlations between ICD codes, we defined the phenome framework by using the PheCODE schema that combines ≥ 1 related ICD codes into distinct outcome groups³. For a given phenotype, the case group included patients recorded as having the specific phecode that most closely related to the etiology of the disease, and the control group was defined on the basis of the absence of the phecode. Participants with a disease code that was related to 1 of the examined case group were also excluded from the control group⁴. Male- and female-specific PheCODEs were generated based on Phemap v1.2 using phecodeX (<https://github.com/PheWAS/PhecodeX>).

Methods in the manuscript (lines 845-847):

“The PheCODE schema combines correlated ICD codes into a distinct code based on the Phemap v1.2 (https://phewascatalog.org/phecodes_icd10) and excludes patients with related diseases from the corresponding control groups⁶⁷.”

- how much sample overlap existed between exposure and outcome datasets in the MR analyses? Could this introduce bias?

Response:

Thank you for this comment. We have compared the cases and samples between the exposure (BMFF GWAS) and each outcome GWAS. The numbers are as follows:

	Exposure-BMFF (PMID: 39747859)				Outcome-Osteoporosis
	Femoral head	Total hip	Diaphysis	Spine	GCST90038656
Sample N	38581	38394	37513	41204	484598
Case n	416	433	414	438	7751

	Exposure-BMFF (PMID: 39747859)				Outcome-Fractures
	Femoral head	Total hip	Diaphysis	Spine	GCST90038703
Sample N	38581	38394	37513	41204	484598
Case n	736	737	714	783	8844

	Exposure-BMFF (PMID: 39747859)				Outcome-Type 2 diabetes
	Femoral head	Total hip	Diaphysis	Spine	GCST90018926
Sample N	38581	38394	37513	41204	490089
Case n	506	494	483	538	38841

As shown above, <5% case and sample overlap was found between exposure-BMFF GWAS and each outcome GWAS. We understand that sample overlap can introduce bias in MR analyses, particularly violating the independence assumption in two-sample MR. However, at <5% overlap, the bias is minimal, as simulations suggest lower overlaps (e.g., <10%) have less pronounced effects on false discovery proportions (FDP), with corrections bringing FDP back to nominal levels.

We have added this as a limitation in the manuscript (lines 733-735):

“In addition, although sample overlap between exposure and outcome GWAS was minimal (<5%), residual bias cannot be entirely ruled out. We seek to validate our MR findings in the independent dataset once these data are available.”

- please check whether the GWAS on osteoporosis and fractures included EUR ancestry participants only.

Response:

Thank you for this comment. We have checked that GWAS on osteoporosis and fractures included EUR ancestry participants only.

- while simpler PRS are more interpretable for causal inference, the inclusion of correlated SNPs reduces this advantage. If the goal was instead to characterise the broader phenotypic profile of innate BMFF predisposition through the PRS-PheWAS, wouldn't more advanced

methods (e.g., PRS-CS, LDpred, SBayesRC) be more suitable?

Response:

We appreciate the reviewer's insightful suggestion regarding advanced PRS methods. We understand that PRS-CS utilizes Bayesian regression framework and is distinct from our method by placing a continuous shrinkage (CS) prior on SNP effect sizes. This is robust to varying genetic architectures and enables multivariate modelling of local LD patterns.

Our PRS-PheWAS aimed to identify phenotypic associations driven by strong genetic variants rather than maximizing polygenic prediction. PRS-CT constructions align with MR assumptions by minimizing weak instrument bias and LD-related confounding. PRS-CS (which require LD reference panels and extensive tuning) was beyond the scope of this causal inference aim.

We agree that advanced PRS methods would be valuable for future work characterizing the full polygenic architecture of BMFF. Therefore, we have discussed this limitation in the manuscript (lines 712-717):

"Fifthly, we generated PRSs using PRS-CT (clumping + p value thresholding: to generate weighted PRS based on SNP associated with BMFF) because it is more interpretable for causal inference. However, the inclusion of correlated SNPs reduces this advantage. The PRS-CS method, which utilizes a Bayesian regression framework by placing a continuous shrinkage (CS) prior on SNP effect sizes⁶¹, would be valuable for future work characterizing the full polygenic architecture of BMFF."

Minor

- lines #236-237: please include the scaling used for BMFF in its GWAS when interpreting the effect sizes. The reported coefficients (e.g., 0.189 to 0.381) are otherwise uninterpretable. Also, are these log-odds?

Response:

Many thanks for the comment. We have revised manuscript (lines 237-238): *"Positive associations were observed, with β coefficients (log OR) ranging from 0.189 to 0.381 ($P < 2 \times 10^{-16}$)."*

- what LD reference panel was used when assessing LD R2 among SNPs?

Response:

Many thanks for the comment. 1000 Genomes Project Phase 3 was used as LD reference panel.

1) SNPs (BMFF meta-GWAS) identified by FUMA:

'In FUMA, this task is automated by using pairwise LD (r^2) of SNPs in the reference panel (1000 genomes project phase 3) pre-computed by PLINK, resulting in a list of independent significant SNPs, lead SNPs and genomic risk loci based on the GWAS input file.'

(<https://fuma.ctglab.nl/tutorial>; <https://www.ncbi.nlm.nih.gov/pmc/articles/PMC5705698/>)

2) LDLinkR (exclude LD proxies associated with BMD/T2D):

'LDlink relies on the 1000 Genomes Project Phase 3 as its reference panel.'

(<https://dceg.cancer.gov/tools/analysis/ldlink>;

<https://academic.oup.com/bioinformatics/article/31/21/3555/195027>)

We have added this reference in manuscript (lines 809-811):

"LD proxies ($r^2 > 0.6$) for significant SNPs passing quality control were generated using the LDproxy function in the LDlinkR package (<https://github.com/CBIIT/LDlinkR>, 1000 Genomes Project Phase 3 as LD reference panel)."

- the term polygenic risk score (PRS) is technically correct but now often connotes genome-wide, highly polygenic approaches. Since your score is based on genome-wide significant hits, please clarify this in the abstract and main text.

Response:

Many thanks for the comment. We have revised 'PRS' term in the abstract (line 54) and main text (lines 795-798):

- Abstract: *"We then establish polygenic risk scores (PRSs) based on genome-wide significant susceptibility SNPs associated with BMFF and use PRS-PheWAS and Mendelian Randomization to explore potential causal association between BMFF and disease outcomes."*
- Main text: *"the PRS for each bone region was constructed by adding up the number of BMFF-increasing alleles for each SNP, weighted for the SNP effect size on BMFF level (beta coefficients from meta-GWAS)."*

- a few suggestions to improve readability of your tables: (i) remove log-odds if you already present ORs and 95% CIs, (ii) reduce decimals to 1-2 to avoid pseudo-precision, (iii) format sample sizes with commas (e.g., 10,000)

Response:

We appreciate the reviewer's suggestions for improving table readability. We have made the following changes, but retained the current format for:

- Log-odds (beta):

We included both log-odds (beta) and ORs to accommodate different analytical

interpretations - log-odds (beta) provide linear scale effects, while ORs offer clinical interpretability. This aligns with Nature Communications' reporting standards.

- Decimal:

The 3 decimal places were preserved to ensure consistency because some Egger intercepts are very small (e.g., 0.002). In Supplementary Tables 14-15, we report r^2 values to 4 decimal places so that it is clear which SNPs are included in the LD clump $r^2 < 0.001$. In Supplementary Table 18, we report C-statistics to 4 decimal places so that it is clear how these differ between PRS1 vs PRS2. In other cases where values are typically < 0.001 , we report data in Scientific notation.

- Sample Size:

We have comma-separated sample values.

We are happy to adjust these in any final publication versions per journal style requirement.

- from your GWAS publication, it appears you ran both BMI-adjusted and unadjusted GWAS. Which version was used for score creation and MR? If the BMI-adjusted version was used, could collider bias influence your results?

Response:

We thank the reviewer for this valuable comment. BMI-adjusted SNPs were used. We understand the concerns that if BMI-adjusted GWAS is used for MR, collider bias may arise if BMI is a mediator (e.g., exposure \rightarrow BMI \rightarrow outcome).

To assess the impact of BMI adjustment on the BMFF trait associations, we compared the meta-GWAS results with and without BMI adjustment for the four bone regions. Our analysis did not find any distinct discrepancies in the results. The direction of the effect estimates was consistent with and without BMI adjustment, with median beta differences around 1.6×10^{-5} . Please find details in Supplementary Tables 15 and 17 of our meta-GWAS paper (Xu et al. 2025).

To address this comment, we have compared MR results (LD clump $r^2 < 0.001$) based on IVs from meta-GWAS with and without BMI adjustment to assess whether BMI adjustment introduced collider bias in the MR. Our findings are reported in the updated Results section (lines 373-379) as follows:

“Finally, to assess whether BMI adjustment introduced collider bias in the MR, we compared MR results ($r^2 < 0.001$) based on IVs from meta-GWAS with and without BMI adjustment¹⁵. We found no distinct discrepancies in the comparisons (Supplementary Tables 33-35), with the effect estimates remaining directionally consistent between BMI-adjusted and unadjusted analyses (Osteoporosis: median OR difference = 0.027; Fractures: median OR difference = 0.022; T2D: median OR difference = 0.004). Thus, BMI adjustment does not significantly alter the MR results.”

- for datasets obtained via MRC IEU, please consider including references to the original GWAS studies.

Response:

Many thanks for the comment. We have cited references for the osteoporosis GWAS⁷⁶ (ebi-a-GCST90038656), for the fractures GWAS⁷⁶ (ebi-a-GCST90038703), and for the type 2 diabetes GWAS⁷⁷ (ebi-a-GCST90018926); this information is now in lines 903-904 of the revised manuscript.

- please check STable 17 for remaining notes that should be removed.

Response:

Many thanks for the comment. The remaining notes for Supplementary Table 17 have been removed.

Reviewer #3 (Remarks to the Author):

This manuscript is a comprehensive analysis investigating diseases associated with bone marrow adipose tissue (BMAT). This analysis builds on previously published work that used deep learning to measure BMFF of 4 sites (femoral head, total hip, femoral diaphysis, spine) of >44,00 participants in the UK Biobank and genome-wide association studies. The authors now examine BMAT associations using phenome-wide association studies (PheWAS) through different methods: an Observational-PheWAS, a polygenic risk score (PRS)-PheWAS, and Mendelian Randomization.

In the Observational-PheWAS, BMFF was associated with 47 incident diseases across 12 disease categories, which included osteoporosis, fracture, type 2 diabetes, and also diseases not previously linked to BMAT, such as cardiovascular diseases, several cancers, and other conditions. The PRS-PheWAS and Mendelian Randomization analyses provide additional evidence and causality for a relationship between BMAT and osteoporosis, fractures, and T2D.

There is growing appreciation for the importance of BMAT, a unique and dynamic fat depot, that has significant implications on skeletal and systemic metabolism and nutrition. This manuscript addresses a significant gap in the comprehensive and large-scale evaluation of the physiologic and pathologic roles of BMAT. Significant strengths include methods using multiple approaches to evaluate the relationship of BMAT to PheWAS (Observational-PheWAS and PRS-PheWAS), Mendelian Randomization to infer causality, and a thoughtful approach to specifically evaluate osteoporosis diagnoses, fracture, and findings independent of BMD. Another significant strength is the large sample size using a gold standard method to analyze BMAT and analyzing 4 skeletal sites, which is valuable given site-specific variations in BMAT. This is a well written manuscript by leaders in the field. My concerns are noted below.

Response:

Thank you for your positive comments about our study and for recognising our contributions to this field; we truly appreciate your feedback. We have addressed your insightful points as described below.

1. Confounding: The Obs-PheWAS analyses were adjusted for age, sex, and BMI at imaging. The PRS-PheWAS analyses were adjusted for age, sex, assessment centre, and the first 10 genetic principal components. There could be confounding variables that may be influencing associations. For example, glucocorticoid use increases BMAT and is associated with COPD. It may not be possible to account for or adjust for all the various factors, but confounding

factors should be considered, especially for BMAT associations with unexpected diseases.

Response:

We agree that variables, such as medication use, may confound the BMFF-disease associations. It would be relatively straightforward to account for these covariables if we were assessing the relationships between BMFF and a specific disease, because the disease-influencing covariables can be more readily identified; however, this is far more complex in the context of PheWAS because it identifies a broad range of very different diseases, each of which can show distinct relationships with medication use and other covariables. Therefore, identifying and controlling for these co-variables, for all PheWAS outcomes, is extremely challenging. Moreover, UK Biobank does not have all data for medication history; hence, it would not be possible to control for medication use in a systematic manner. Nevertheless, we agree that this is an important point, and therefore we have highlighted it as a limitation in the discussion of our updated manuscript (Lines 700-705), as follows:

“Thirdly, medication use may influence some of the observed BMFF-disease associations. For example, glucocorticoids are used to manage COPD and they also increase BMFF². However, UKBB participant medication history is mostly from self-reported data, with primary care records being available for only a subset of participants; thus, it is not possible to systematically control for use of medications that may influence the observed associations between BMFF and disease outcomes.”

2. Phenome: Given the importance of ICD codes for mapping the phenome, it would be important to provide additional information and note important limitations. The authors note that disease outcomes were defined based on national medical records (inpatient hospital episode records, cancer registry, and death registry). Would this miss ambulatory diagnoses? This may affect the case detection for osteoporosis and diabetes, and other diseases managed in the ambulatory setting.

Response:

Thank you for your insightful comment. You raise an important point about the possible under-ascertainment of disease outcomes such as osteoporosis and diabetes that are often managed in ambulatory settings. As above for point 1, the lack of comprehensive primary care data prevents systematic consideration of these diagnoses in a primary care setting. Thus, we have also addressed this as a limitation in the Discussion (lines 707-710) as follows:

“Fourthly, we defined PheCODES based on national medical records (inpatient hospital episode records, cancer registry, and death registry) in UKBB, which did not include diagnoses

from primary care. This may result in incomplete case ascertainment for conditions predominantly diagnosed and managed in the primary care setting.”

3. Site-specific BMAT: Given the site-specific nature of these findings, it would be helpful if the authors could briefly elaborate on the biological significance and how these findings could inform our understanding of regulated and constitutive BMAT.

Response:

We agree on the importance of the site-specific nature of our findings and highlighted this in the Discussion, particularly in the section “Site and sex differences in BMFF-associated diseases”. Although we feel our existing text addresses the biological significance, we agree that it would be helpful to put the site-specific findings in the context of regulated and constitutive BMAT. Therefore, we have updated the text in this section (lines 632-648 of revised manuscript) to read:

“These findings highlight the critical importance of analyzing BMFF at multiple anatomical locations and underscore the concept that the pathophysiological functions of BMAT are skeletal site dependent^{2,20,21}. Indeed, BMAT has been proposed to exist in two broad subtypes, regulated BMAT (rBMAT) and constitutive BMAT (cBMAT), which differ in anatomical location and in their molecular and functional characteristics^{2,20,21}. Thus, cBMAT forms earlier in development, contains densely packed adipocytes, predominates at distal skeletal sites, and is less changeable in response to environmental and pathophysiological variables. In contrast, rBMAT forms later in development, features adipocytes interspersed with hematopoietic red marrow, predominates in the axial skeleton and more-proximal regions of the limbs, and more readily increases or decreases in response to varying environmental stimuli and pathophysiological states. These characteristics are broadly consistent with our observations, with spine BMFF corresponding to rBMAT and femoral BMFF being more cBMAT-like. However, some disease associations are even stronger for femoral BMFF than for spine BMFF, suggesting that cBMAT is not always more inert than rBMAT. The three femoral sites also differ to one another, suggesting that this binary cBMAT/rBMAT classification is an oversimplification. It will be important for future studies to further dissect these site-specific differences on molecular and functional levels.”

4. Spine BMAT and skeletal outcomes: There was not an observed causal association for spine BMFF and osteoporosis. Prior studies found that spine BMFF had a robust inverse association with spine BMD. Based on the earlier concerns, if there was an under-detection of osteoporosis at the spine and many vertebral compression fractures are asymptomatic, this may affect the ability to detect this relationship. The authors nicely summarize these

limitations later in the discussion, but it may be more helpful to include these earlier when discussing the interpretation of this finding.

Response:

We agree that this is an important, complex point to highlight. We feel the full explanation is best placed in the Discussion, but we have updated this part of the results (Lines 331-333) with the following text:

“As outlined in the Discussion, the lack of causality for spine BMFF is surprising and could have several explanations, including osteoporosis diagnoses in UKBB being driven more by bone loss in the hip than in the spine.”

We think this helps the reader to understand that additional explanations and interpretation are forthcoming, without disrupting the flow of the results.

References cited in response to reviewers:

Ambrosi, T. H., A. Scialdone, A. Graja, S. Gohlke, A. M. Jank, C. Bocian, . . . T. J. Schulz (2017). "Adipocyte Accumulation in the Bone Marrow during Obesity and Aging Impairs Stem Cell-Based Hematopoietic and Bone Regeneration." Cell Stem Cell **20**(6): 771-784 e776. doi:10.1016/j.stem.2017.02.009

Ambrosi, T. H., S. Taheri, K. Chen, R. Sinha, Y. Wang, E. J. Hunt, . . . C. K. F. Chan (2025). "Human skeletal development and regeneration are shaped by functional diversity of stem cells across skeletal sites." Cell Stem Cell **32**(5): 811-823.e811. doi:<https://doi.org/10.1016/j.stem.2025.02.013>

Craft, C. S., Z. Li, O. A. MacDougald and E. L. Scheller (2018). "Molecular differences between subtypes of bone marrow adipocytes." Current Molecular Biology Reports **4**(1): 16-23.

Francis, C. M., M. E. Futschik, J. Huang, W. Bai, M. Sargurupremraj, A. Teumer, . . . P. M. Matthews (2022). "Genome-wide associations of aortic distensibility suggest causality for aortic aneurysms and brain white matter hyperintensities." Nat Commun **13**(1): 4505. doi:10.1038/s41467-022-32219-x

Gimble, J. M., C. E. Robinson, X. Wu and K. A. Kelly (1996). "The function of adipocytes in the bone marrow stroma: an update." Bone **19**(5): 421-428.

Gorman, B. R., S. G. Ji, M. Francis, A. K. Sendamarai, Y. Shi, P. Devineni, . . . S. Pyarajan (2024). "Multi-ancestry GWAS meta-analyses of lung cancer reveal susceptibility loci and elucidate smoking-independent genetic risk." Nat Commun **15**(1): 8629. doi:10.1038/s41467-024-52129-4

Jansen, I. E., J. E. Savage, K. Watanabe, J. Bryois, D. M. Williams, S. Steinberg, . . . D. Posthuma (2019). "Genome-wide meta-analysis identifies new loci and functional pathways influencing Alzheimer's disease risk." Nat Genet **51**(3): 404-413. doi:10.1038/s41588-018-0311-9

Jansen, P. R., M. Nagel, K. Watanabe, Y. Wei, J. E. Savage, C. A. de Leeuw, . . . D. Posthuma (2020). "Genome-wide meta-analysis of brain volume identifies genomic loci and genes shared with intelligence." Nat Commun **11**(1): 5606. doi:10.1038/s41467-020-19378-5

Karampinos, D. C., S. Ruschke, M. Dieckmeyer, M. Diefenbach, D. Franz, A. S. Gersing, . . . T. Baum (2018). "Quantitative MRI and spectroscopy of bone marrow." J Magn Reson Imaging **47**(2): 332-353. doi:10.1002/jmri.25769

Khan, A., N. Shang, J. G. Nestor, C. Weng, G. Hripcsak, P. C. Harris, . . . K. Kiryluk (2023). "Polygenic risk alters the penetrance of monogenic kidney disease." Nat Commun **14**(1): 8318. doi:10.1038/s41467-023-43878-9

Korologou-Linden, R., L. Bhatta, B. M. Brumpton, L. D. Howe, L. A. C. Millard, K. Kolaric, . . . N. M. Davies (2022). "The causes and consequences of Alzheimer's disease: phenome-wide evidence from Mendelian randomization." Nat Commun **13**(1): 4726. doi:10.1038/s41467-022-32183-6

Krohn, L., K. Heilbron, C. Blauwendraat, R. H. Reynolds, E. Yu, K. Senkevich, . . . Z. Gan-Or (2022). "Genome-wide association study of REM sleep behavior disorder identifies polygenic risk and brain expression effects." Nat Commun **13**(1): 7496. doi:10.1038/s41467-022-34732-5

Li, L., D. Li, Z. Geng, Z. Huo, Y. Kang, X. Guo, . . . T. Wang (2024). "Causal relationship between bone mineral density and intervertebral disc degeneration: a univariate and multivariable mendelian randomization study." BMC Musculoskelet Disord **25**(1): 517. doi:10.1186/s12891-024-07631-7

Liang, X., L. Liang and Y. Fan (2023). "Two-sample mendelian randomization analysis investigates ambient fine particulate matter's impact on cardiovascular disease development." Sci Rep **13**(1): 20129. doi:10.1038/s41598-023-46816-3

Meunier, P., J. Aaron, C. Edouard and G. Vignon (1971). "Osteoporosis and the replacement of cell populations of the marrow by adipose tissue. A quantitative study of 84 iliac bone biopsies." Clin Orthop Relat Res **80**: 147-154.

Nagel, M., P. R. Jansen, S. Stringer, K. Watanabe, C. A. de Leeuw, J. Bryois, . . . D. Posthuma (2018). "Meta-analysis of genome-wide association studies for neuroticism in 449,484 individuals identifies novel genetic loci and pathways." Nat Genet **50**(7): 920-927. doi:10.1038/s41588-018-0151-7

Rajesh, A. E., A. Olvera-Barrios, A. N. Warwick, Y. Wu, K. V. Stuart, M. I. Biradar, . . . C. Egan (2025). "Machine learning derived retinal pigment score from ophthalmic imaging shows ethnicity is not biology." Nat Commun **16**(1): 60. doi:10.1038/s41467-024-55198-7

Salih, A., M. Ardissino, A. Z. Wagen, A. Bard, L. Szabo, M. Ryten, . . . Z. Raisi-Estabragh (2023). "Genome-Wide Association Study of Pericardial Fat Area in 28 161 UK Biobank Participants." J Am Heart Assoc **12**(21): e030661. doi:10.1161/jaha.123.030661

Scheller, E. L., C. R. Doucette, B. S. Learman, W. P. Cawthorn, S. Khandaker, B. Schell, . . . O. A. MacDougald (2015). "Region-specific variation in the properties of skeletal adipocytes reveals regulated and constitutive marrow adipose tissues." Nat Commun **6**: 7808. doi:10.1038/ncomms8808

Singh, S., A. Choudhury, S. Hazelhurst, N. J. Crowther, P. R. Boua, H. Sorgho, . . . M. Ramsay (2023). "Genome-wide association study meta-analysis of blood pressure traits and hypertension in sub-Saharan African populations: an AWI-Gen study." Nat Commun **14**(1): 8376. doi:10.1038/s41467-023-44079-0

Sproviero, W., L. Winchester, D. Newby, M. Fernandes, L. Shi, S. M. Goodday, . . . A. J. Nevado-Holgado (2021). "High Blood Pressure and Risk of Dementia: A Two-Sample Mendelian Randomization Study in the UK Biobank." Biol Psychiatry **89**(8): 817-824. doi:10.1016/j.biopsych.2020.12.015

Sulston, R. J. and W. P. Cawthorn (2016). "Bone marrow adipose tissue as an endocrine organ: close to the bone?" Hormone Molecular Biology and Clinical Investigation **28**(1): 21-38. doi:10.1515/hmbci-2016-0012

Watanabe, K., E. Taskesen, A. van Bochoven and D. Posthuma (2017). "Functional mapping and annotation of genetic associations with FUMA." Nature Communications **8**(1): 1826. doi:10.1038/s41467-017-01261-5

Xu, W., I. Mesa-Eguiagaray, D. M. Morris, C. Wang, C. D. Gray, S. Sjöström, . . . W. P. Cawthorn (2025). "Deep learning and genome-wide association meta-analyses of bone marrow adiposity in the UK Biobank." Nat Commun **16**(1): 99. doi:<https://doi.org/10.1038/s41467-024-55422-4>

Yu, W., L. Zhong, L. Yao, Y. Wei, T. Gui, Z. Li, . . . L. Qin (2020). "Bone marrow adipogenic lineage precursors (MALPs) promote osteoclastogenesis in bone remodeling and pathologic bone loss." The Journal of Clinical Investigation. doi:10.1172/JCI140214

REVIEWER COMMENTS

Reviewer #2 (Remarks to the Author):

I thank the authors for their detailed replies. From my reading, they have complied with the SAGER guidelines. My remaining, relatively minor comments:

(1) while FUMA uses an initial $r^2 < 0.6$ threshold to define independent signals from GWAS, this is a relatively lenient threshold that results in sets of correlated variants. As the authors rightly note, using such a threshold for instrument selection in Mendelian randomization (MR) leads to spuriously precise estimates and inflated type I error rates. I therefore appreciate the inclusion of results based on more conventional LD thresholds ($r^2 < 0.001$ would also have been appropriate, particularly given the limited resolution of the 1000 Genomes reference panel). However, the rationale for including results based on $r^2 < 0.6$ remains unclear, as the claim that these analyses serve to “demonstrate the impact of IV strength on MR results” seems misleading, as the LD threshold used to define independence does not affect the strength of individual instruments. I suggest moving this sensitivity analysis to the supplementary materials.

Response:

We appreciate the reviewer's insightful and thorough comments on MR analysis.

We have now focussed on the results using LD-clumping with $r^2 < 0.001$ to select a set of independent instruments for each trait. These MR results for $r^2 < 0.001$ are shown as our primary analysis (In-text Tables 5-7). As requested, we now show the sensitivity analysis (results using LD clump threshold of $r^2 < 0.6$) only in the supplementary materials (Supplementary Tables 27-29). We have also updated the abstract to highlight the MR results only based on the $r^2 < 0.001$ results.

To address this comment, we have made the following revisions:

- Methods in the manuscript (lines 895-906):

“We adhered to the three core genetic instrumental variants (IV) assumptions: (1) Relevance: the IV is associated with the exposure of interest; (2) Independence: the IV is independent of any confounders of the exposure-outcome association; and (3) Exclusion Restriction: the IV selected for MR analysis are not associated with the outcome of interest independently of the exposure⁷⁴. To meet the assumptions, we obtained IVs by filtering for BMFF meta-GWAS genome-wide significant SNPs ($P < 5 \times 10^{-8}$), followed by linkage disequilibrium (LD)-based clumping using the ‘TwoSampleMR’ R package with the following parameters: $clump_kb = 10,000$, $clump_r^2$ for two thresholds ($r^2 = 0.001$: primary analysis or $r^2 = 0.6$:

sensitivity analysis), pop = "EUR", and excluded those SNPs associated with BMD or T2D to ensure that any observed association between IVs and outcomes is mediated through the exposure (BMFF), maintaining the validity of causal inference. The strength of genetic IVs was tested with the F-statistic (<https://shiny.cnsgenomics.com/mRnd/>).

- Results in the manuscript (lines 324-363):

"The causal association of BMFF genetic predisposition with osteoporosis is summarized in Table 5. When using the more stringent LD clumping threshold ($r^2 < 0.001$), positive associations with osteoporosis were found for total hip BMFF and diaphysis BMFF based on the inverse-variance weighted (IVW) method and for diaphysis BMFF based on MR-Horse. Moreover, consistent positive associations were detected for all three femoral sites when the less-stringent LD clumping threshold of $r^2 < 0.6$ was applied (Supplementary Table 27). However, no causal association was found in the spine, regardless of the r^2 threshold used."

"The causal association of BMFF genetic liability with fractures is summarized in Table 6. Each of the MR methods found no significant causal associations for any of the four bone regions when using the more stringent LD clumping threshold ($r^2 < 0.001$). However, when the more lenient LD clumping threshold of $r^2 < 0.6$ was applied, IVW and MR-Horse showed positive associations with fractures for all three femoral sites (Supplementary Table 28)."

"The causal association of BMFF genetic liability with T2D is summarized in Table 7. When using the more stringent LD clumping threshold ($r^2 < 0.001$), none of the MR methods found significant causal associations in any of the four bone regions. At the less stringent LD clumping threshold ($r^2 < 0.6$), the findings suggested negative associations with T2D in the total hip and diaphysis, whereas MR-Horse observed a positive association between BMFF genetic liability and T2D in the spine (Supplementary Table 29)."

- Abstract in the manuscript (lines 58-60):

"PRS-PheWAS reveals that genetic predisposition to increased BMFF is positively associated with osteoporosis and fractures. Mendelian Randomization further suggests that increased diaphysis and total hip BMFF are causally associated with osteoporosis."

(2) the exclusion of variants associated with BMD or T2D may have inadvertently removed some of the strongest instruments for MR, and such exclusions do not "ensure that any observed association between IVs and outcomes is mediated through the exposure." Approaches such as Steiger filtering would be more appropriate for assessing the direction of causality in cases where variants are

associated with both exposure and outcome. The authors may wish to consider this in their future work.

Response:

We appreciate the reviewer’s insightful feedback regarding our exclusion of variants associated with BMD or T2D. We opted for this conservative exclusion strategy to minimize horizontal pleiotropy; however, the reviewer raises a valid point about the potential loss of strong instruments for MR and that such exclusions cannot fully guarantee that observed associations are mediated through the exposure. We also agree that Steiger filtering—which assesses the direction of causality by comparing the variance explained in the exposure and outcome—could provide a more-nuanced approach to handling pleiotropic variants, allowing us to retain more instruments while addressing potential reverse causality.

We have now conducted additional analyses to assess whether this exclusion affected the strength of our instrumental variables (IVs) and MR results. We find that excluding BMD- or T2D-associated variants has little impact on IV strength, as reflected by the small number of excluded variants and their F-statistics:

BMD-associated variants:

- Femoral head (rs7537281, F = 84.8), Total hip (rs3020332, F = 68.6), Spine (rs17601876, F = 35.6; rs6684375, F = 44.4).

T2D-associated variants:

- Femoral head (rs76895963, F = 30.0), Total hip (rs11712037, F = 62.6; rs76895963, F = 35.0; rs9379084, F = 31.2), Spine (rs17601876, F = 35.6; rs6684375, F = 44.4).

A direct comparison of IV strength between instrument sets (with and without exclusion) across bone sites showed minimal changes in the number of SNPs and F-statistics (summarized below and now shown in new Supplementary Table 39):

Summary comparison of IV strength between two MR instrument sets:

	IV set (including BMD variants)	IV set (excluding BMD variants)
Femoral head BMFF-Osteoporosis/Fracturs		
Number of SNPs (IVs)	16	15
F-statistics	40.05 (32.79-52.75)	39.64 (32.23-45.96)
Total hip BMFF-Osteoporosis/Fractures		
Number of SNPs (IVs)	29	28
F-statistics	40.14 (34.95-58.41)	39.47 (34.46-53.53)
Diaphysis-Osteoporosis/Fractures		
Number of SNPs (IVs)	19	19
F-statistics	45.68 (37.18-85.01)	45.68 (37.18-85.01)
Spine-Osteoporosis/Fractures		
Number of SNPs (IVs)	36	34

F-statistics	42.87 (32.80-57.52)	42.87 (32.68-57.57)
--------------	---------------------	---------------------

Median (IQR)

	IV set (including T2D variants)	IV set (excluding T2D variants)
Femoral head BMFF-T2D		
Number of SNPs (IVs)	16	15
F-statistics	40.05 (32.79-52.75)	40.47 (33.91-58.81)
Total hip BMFF-T2D		
Number of SNPs (IVs)	29	26
F-statistics	40.14 (34.95-58.41)	41.20 (35.27-56.79)
Diaphysis-T2D		
Number of SNPs (IVs)	19	19
F-statistics	45.68 (37.18-85.01)	45.68 (37.18-85.01)
Spine-T2D		
Number of SNPs (IVs)	37	35
F-statistics	42.87 (32.80-57.52)	42.87 (32.74-57.55)

We have now also conducted the Steiger filtering test¹ to evaluate the direction of causality. Our results (shown in new Supplementary Tables 40-43) support that the genetic instruments explain more variance (snp r²) in the exposure (BMFF) than in the outcomes (osteoporosis, fractures, and T2D), with all tests indicating correct causal direction (all Steiger p-values significant).

Steiger filtering results (Supplementary Table 40):

exposure	outcome	snp_r2.exposure	snp_r2.outcome	correct_causal_direction	steiger_pval
BMFF-Osteoporosis					
femoral head BMFF	osteoporosis	2.23E-02	4.41E-04	TRUE	4.19E-132
total hip BMFF	osteoporosis	4.05E-02	5.86E-04	TRUE	3.67E-252
diaphysis BMFF	osteoporosis	3.42E-02	5.38E-04	TRUE	2.21E-205
spine BMFF	osteoporosis	4.38E-02	2.28E-04	TRUE	0
BMFF-Fractures					
femoral head BMFF	fracture	2.23E-02	5.77E-05	TRUE	1.93E-160
total hip BMFF	fracture	4.05E-02	7.74E-05	TRUE	7.24E-297
diaphysis BMFF	fracture	3.42E-02	6.33E-05	TRUE	5.84E-245
spine BMFF	fracture	4.38E-02	7.46E-05	TRUE	0
BMFF-T2D					
femoral head BMFF	T2D	2.23E-02	5.38E-04	TRUE	7.99E-128
total hip BMFF	T2D	4.05E-02	9.55E-04	TRUE	4.55E-234
diaphysis BMFF	T2D	3.53E-02	2.31E-04	TRUE	4.56E-234
spine BMFF	T2D	4.38E-02	2.83E-04	TRUE	0

Thus, we further tested whether inclusion of these variants would influence the MR estimates. We found that their inclusion did not alter the significant associations reported in our original MR analyses: increased diaphysis and total hip BMFF remained causally associated with osteoporosis. These findings are now shown in Supplementary Tables 41-43:

BMFF-Osteoporosis MR-steiger filtering test and MR-exclude BMD-associated SNPs cross-comparison (Supplementary Table 41):

Bone region	Method	Beta_steiger	SE_steiger	P-value_steiger	FDR-Q_steiger	Beta_excludeBMD	SE_excludeBMD	P-value_excludeBMD	FDR-Q_excludeBMD	Beta_diff	FDR-Q_diff
Femoral head	MR Horse	0.536	0.292	0.066	0.159	0.435	0.267	0.103	0.207	0.101	-0.047
Femoral head	IVW	0.406	0.244	0.096	0.193	0.222	0.225	0.324	0.409	0.184	-0.217
Femoral head	MR Egger	2.553	0.775	0.005	0.018	1.929	0.823	0.036	0.095	0.624	-0.077
Femoral head	Weighted median	0.211	0.176	0.230	0.328	0.070	0.175	0.688	0.786	0.141	-0.458
Femoral head	Simple mode	-0.060	0.362	0.871	0.918	0.006	0.332	0.986	0.986	-0.066	-0.068
Femoral head	Weighted mode	-0.060	0.388	0.880	0.918	-0.007	0.366	0.984	0.986	-0.052	-0.068
Total hip	MR Horse	0.218	0.147	0.138	0.255	0.180	0.144	0.211	0.305	0.038	-0.051
Total hip	IVW	0.609	0.170	0.000	0.002	0.559	0.171	0.001	0.005	0.049	-0.003
Total hip	MR Egger	1.258	0.508	0.020	0.060	1.253	0.497	0.018	0.062	0.005	-0.003
Total hip	Weighted median	0.214	0.126	0.089	0.193	0.208	0.121	0.085	0.186	0.006	0.007
Total hip	Simple mode	-0.105	0.166	0.531	0.638	-0.110	0.164	0.507	0.608	0.006	0.029
Total hip	Weighted mode	-0.014	0.167	0.932	0.932	-0.009	0.177	0.958	0.986	-0.005	-0.054
Diaphysis	MR Horse	0.775	0.152	3.42E-07	2.74E-06	0.775	0.152	3.42E-07	2.74E-06	0.000	0.000
Diaphysis	IVW	0.841	0.159	1.23E-07	1.47E-06	0.841	0.159	1.23E-07	1.47E-06	0.000	0.000
Diaphysis	MR Egger	1.221	0.494	0.024	0.065	1.221	0.494	0.024	0.073	0.000	-0.008
Diaphysis	Weighted median	0.914	0.127	6.71E-13	1.61E-11	0.914	0.127	6.71E-13	1.61E-11	0.000	0.000
Diaphysis	Simple mode	0.812	0.221	0.002	0.007	0.812	0.221	0.002	0.007	0.000	0.000
Diaphysis	Weighted mode	0.898	0.221	0.001	0.004	0.898	0.221	0.001	0.004	0.000	-0.001
Spine	MR Horse	-0.086	0.084	0.306	0.408	-0.110	0.080	0.169	0.271	0.024	0.137
Spine	IVW	-0.040	0.107	0.707	0.808	-0.123	0.082	0.131	0.243	0.083	0.566
Spine	MR Egger	-0.562	0.404	0.173	0.296	-0.615	0.303	0.051	0.121	0.052	0.175
Spine	Weighted median	-0.120	0.101	0.232	0.328	-0.141	0.103	0.169	0.271	0.021	0.057
Spine	Simple mode	-0.247	0.275	0.375	0.473	-0.288	0.287	0.323	0.409	0.041	0.064
Spine	Weighted mode	-0.310	0.253	0.227	0.328	-0.350	0.278	0.216	0.305	0.040	0.022

BMFF-Fractures MR-steiger filtering test and MR-exclude BMD-associated SNPs cross-comparison (Supplementary Table 42):

Bone region	Method	Beta_steiger	SE_steiger	P-value_steiger	FDR-Q_steiger	Beta_excludeBMD	SE_excludeBMD	P-value_excludeBMD	FDR-Q_excludeBMD	Beta_diff	FDR-Q_diff
Femoral head	MR Horse	0.218	0.100	0.029	0.150	0.188	0.109	0.085	0.281	0.030	-0.131
Femoral head	IVW	0.237	0.093	0.011	0.150	0.214	0.101	0.033	0.281	0.022	-0.131
Femoral head	MR Egger	0.726	0.347	0.055	0.188	0.705	0.405	0.105	0.281	0.021	-0.092
Femoral head	Weighted median	0.285	0.128	0.025	0.150	0.252	0.130	0.053	0.281	0.034	-0.131
Femoral head	Simple mode	0.331	0.221	0.155	0.311	0.303	0.236	0.219	0.345	0.028	-0.034
Femoral head	Weighted mode	0.345	0.223	0.142	0.311	0.308	0.245	0.228	0.345	0.037	-0.034
Total hip	MR Horse	0.140	0.065	0.031	0.150	0.114	0.067	0.089	0.281	0.026	-0.131
Total hip	IVW	0.134	0.064	0.037	0.150	0.106	0.062	0.087	0.281	0.028	-0.131
Total hip	MR Egger	0.235	0.198	0.246	0.368	0.233	0.186	0.222	0.345	0.003	0.024
Total hip	Weighted median	0.204	0.087	0.019	0.150	0.200	0.089	0.025	0.281	0.004	-0.131
Total hip	Simple mode	0.189	0.154	0.232	0.368	0.185	0.150	0.229	0.345	0.004	0.024
Total hip	Weighted mode	0.216	0.122	0.087	0.262	0.218	0.126	0.096	0.281	-0.001	-0.019
Diaphysis	MR Horse	0.086	0.079	0.276	0.390	0.086	0.079	0.276	0.390	0.000	0.000
Diaphysis	IVW	0.111	0.077	0.148	0.345	0.111	0.077	0.148	0.345	0.000	0.000
Diaphysis	MR Egger	0.398	0.232	0.105	0.281	0.398	0.232	0.105	0.281	0.000	0.000
Diaphysis	Weighted median	0.127	0.094	0.176	0.345	0.127	0.094	0.176	0.345	0.000	0.000
Diaphysis	Simple mode	0.101	0.159	0.535	0.713	0.101	0.159	0.535	0.713	0.000	0.000
Diaphysis	Weighted mode	0.151	0.122	0.230	0.345	0.151	0.122	0.230	0.345	0.000	0.000
Spine	MR Horse	0.018	0.061	0.768	0.877	-0.008	0.061	0.896	0.910	0.026	-0.033
Spine	IVW	0.008	0.056	0.882	0.882	-0.020	0.057	0.731	0.877	0.028	0.005
Spine	MR Egger	-0.084	0.215	0.700	0.840	-0.028	0.219	0.898	0.910	-0.055	-0.070
Spine	Weighted median	0.015	0.081	0.856	0.882	0.009	0.082	0.910	0.910	0.006	-0.028
Spine	Simple mode	-0.050	0.202	0.804	0.877	-0.052	0.187	0.785	0.897	0.001	-0.020
Spine	Weighted mode	0.084	0.167	0.620	0.783	0.076	0.177	0.670	0.847	0.007	-0.064

BMFF-T2D MR-steiger filtering test and MR-exclude T2D-associated SNPs cross-comparison (Supplementary Table 43):

Bone region	Method	Beta_steiger	SE_steiger	P-value_steiger	FDR-Q_steiger	Beta_excludeT2D	SE_excludeT2D	P-value_excludeT2D	FDR-Q_excludeT2D	Beta_diff	FDR-Q_diff
Femoral head	MR Horse	0.068	0.06	0.257	0.386	0.072	0.054	0.182	0.397	-0.004	-0.011
Femoral head	IVW	-0.035	0.157	0.822	0.858	0.062	0.056	0.269	0.497	-0.098	0.361
Femoral head	MR Egger	-0.927	0.579	0.132	0.263	-0.014	0.245	0.954	0.978	-0.912	-0.715
Femoral head	Weighted median	0.093	0.061	0.130	0.263	0.094	0.064	0.142	0.341	-0.001	-0.078
Femoral head	Simple mode	0.123	0.094	0.211	0.364	0.111	0.113	0.343	0.549	0.011	-0.185
Femoral head	Weighted mode	0.123	0.101	0.246	0.386	0.119	0.120	0.339	0.549	0.003	-0.163
Total hip	MR Horse	-0.076	0.041	0.064	0.263	-0.072	0.035	0.040	0.318	-0.004	-0.055
Total hip	IVW	-0.209	0.101	0.039	0.263	-0.068	0.036	0.062	0.318	-0.141	-0.055
Total hip	MR Egger	-0.565	0.301	0.071	0.263	-0.092	0.112	0.421	0.594	-0.473	-0.331
Total hip	Weighted median	-0.033	0.040	0.412	0.582	-0.029	0.040	0.463	0.617	-0.003	-0.035
Total hip	Simple mode	-0.028	0.058	0.630	0.753	-0.022	0.068	0.754	0.862	-0.007	-0.109
Total hip	Weighted mode	-0.020	0.046	0.659	0.753	-0.018	0.058	0.753	0.862	-0.002	-0.109
Diaphysis	MR Horse	-0.057	0.035	0.103	0.263	-0.057	0.035	0.103	0.318	0.000	-0.055
Diaphysis	IVW	-0.127	0.063	0.042	0.263	-0.127	0.063	0.042	0.318	0.000	-0.055
Diaphysis	MR Egger	-0.362	0.190	0.074	0.263	-0.362	0.190	0.074	0.318	0.000	-0.055
Diaphysis	Weighted median	-0.067	0.041	0.098	0.263	-0.067	0.041	0.098	0.318	0.000	-0.055
Diaphysis	Simple mode	-0.118	0.064	0.081	0.263	-0.118	0.064	0.081	0.318	0.000	-0.055
Diaphysis	Weighted mode	-0.092	0.054	0.106	0.263	-0.092	0.054	0.106	0.318	0.000	-0.055
Spine	MR Horse	0.031	0.042	0.460	0.614	0.034	0.041	0.407	0.594	-0.003	0.020
Spine	IVW	-0.012	0.050	0.816	0.858	0.006	0.045	0.887	0.968	-0.018	-0.110
Spine	MR Egger	-0.0001	0.200	0.9995	0.9995	-0.005	0.179	0.978	0.978	0.005	0.022
Spine	Weighted median	0.062	0.041	0.131	0.263	0.063	0.042	0.137	0.341	-0.001	-0.078
Spine	Simple mode	0.040	0.081	0.624	0.753	0.038	0.082	0.644	0.813	0.002	-0.060
Spine	Weighted mode	0.076	0.060	0.212	0.364	0.078	0.062	0.219	0.438	-0.002	-0.074

To address this comment, we have made the following revisions:

Discussion in the manuscript (lines 734-740):

...a potential limitation of our conservative exclusion of BMD- and T2D-associated variants is that, while this approach reduces the risk of horizontal pleiotropy, it may also lead to a reduction in IVs strength. However, excluding these variants in our MR analyses had minimal impact on overall IVs strength (Supplementary Table 39). Moreover, Steiger filtering⁶³ confirmed correct causal directions for all variants, including those associated with both exposure and outcome (Supplementary Tables 40-43).

(3) please clarify the reference to PhecodeX, as this represents a different phecode map than v1.2. Additionally, the method by which phecodes were derived remains somewhat unclear. The recommended practice is to define cases as individuals with two or more ICD codes mapping to the same phecode: either two identical codes or two distinct codes that map to the same phecode. Individuals with only one relevant ICD code could then be coded as missing (NA) for that phecode. Despite a previous request for clarification, it remains uncertain whether this approach was followed. Moreover, did the authors use individual-level records (e.g., HES) which capture all ICD code occurrences, or relied on summary diagnosis fields (e.g., UKB Field 41270), which only include distinct diagnosis codes, thereby missing repeated instances of the same ICD code? I do not suggest reanalysing the data if summary fields were used, but it is essential that the methods are clearly described their methods to ensure transparency and reproducibility, and so potential limitations in the current approach can be acknowledged.

Response:

Thank you for this comment. We apologize for the confusion regarding the reference to *PhecodeX*; this was an inadvertent reference, and PheCODEs were generated based on Phemap v1.2 (https://phewascatalog.org/phecodes_icd10); this is stated in the Methods (line 854 of our manuscript). To derive PheCODEs, we used individual-level ICD code records (e.g., HES data) to construct phecount. Regarding case definitions, we followed a “minimum code count = 1” approach, (individuals with ≥ 1 ICD code mapping to a given PheCODE). We acknowledge that this differs from the approach (≥ 2 ICD mapping) recommended, but we followed this definition to maximize case inclusion and statistical power, consistent with our previously published PheWAS studies²⁻⁶.

To clarify this, we have updated the Methods (lines 853-860):

“The PheCODE schema combines correlated ICD codes into a distinct code based on the Phemap v1.2 (https://phewascatalog.org/phecodes_icd10) and excludes patients with related

diseases from the corresponding control groups⁶⁸. Herein, we defined cases based on the presence of at least one ICD code mapping to each PheCODE, consistent with our previous publications⁶⁹⁻⁷³. We translated the ICD codes into PheCODE groups, which included 1808 PheCODEs classified into 17 disease categories^{70,74-76}. Further details of the PheCODE schema and application are provided in the Supplementary Information (Supplementary Note 2).”

We also recognize the limitation of ≥ 1 ICD mapping and so have further detailed this in the Supplementary Information (Supplementary Note 2), which includes further details of the PheCODE generation process and references of our previously published PheWAS for PheCODE schema information. These updates to Supplementary Note 2 are as follows:

Supplementary note 2:

“Disease outcomes were defined based on the PheCODE schema (with 10,750 unique ICD-10 codes and 3,113 ICD-9 codes) using national medical records (individual level records: inpatient hospital episode records (HES), cancer registry, and death registry) until March 31, 2023. The PheCODE schema combines correlated ICD codes into a distinct code based on the Phemap v1.2 (https://phewascatalog.org/phecodes_icd10) and excludes patients with related diseases from the corresponding control groups. To account for the correlations between ICD codes, we defined the phenome framework by using the PheCODE schema that combines ≥ 1 related ICD codes into distinct outcome groups. For a given phenotype, the case group included patients recorded as having the specific phecode that most closely related to the etiology of the disease, and the control group was defined on the basis of the absence of the phecode. Participants with a disease code that was related to 1 of the examined case group were also excluded from the control group. Male- and female-specific PheCODEs were generated based on Phemap v1.2.

Limitation of ≥ 1 ICD code mapping to each PheCODE:

In this study, cases were defined based on the presence of at least one ICD code mapping to each PheCODE, consistent with our previous publications⁵⁻⁹. While this approach increases sensitivity and preserves statistical power, it has several limitations. First, it may introduce misclassification and elevate the risk of false positives, as a single ICD code can reflect provisional, erroneous, or rule-out diagnoses. Second, phenotype specificity may be reduced, since individuals with only one relevant code may not truly meet diagnostic criteria. Third, this approach lacks confirmation of chronicity or recurrence, which could be better captured by requiring multiple ICD code occurrences.”

Reviewer #3 (Remarks to the Author):

The authors have thoroughly addressed my concerns in the response.

Response:

Thank you for your thoughtful review of our work.

Reviewer #1:

Please note that Reviewer #1 was not available to review, but we asked Reviewer #3 to check your responses to Reviewer #1's comments.

References cited in response to reviewers:

1. Hemani, G., Tilling, K. & Davey Smith, G. Orienting the causal relationship between imprecisely measured traits using GWAS summary data. *PLoS Genet* **13**, e1007081 (2017).
2. Li, X. *et al.* MR-PheWAS: exploring the causal effect of SUA level on multiple disease outcomes by using genetic instruments in UK Biobank. *Ann Rheum Dis* (2018).
3. Meng, X. *et al.* Phenome-wide Mendelian-randomization study of genetically determined vitamin D on multiple health outcomes using the UK Biobank study. *Int J Epidemiol* **48**, 1425-1434 (2019).
4. Zhang, X. *et al.* Phenome-wide association study (PheWAS) of colorectal cancer risk SNP effects on health outcomes in UK Biobank. *Br J Cancer* **126**, 822-830 (2022).
5. Wang, L. *et al.* Phenome-wide association study of genetically predicted B vitamins and homocysteine biomarkers with multiple health and disease outcomes: analysis of the UK Biobank. *Am J Clin Nutr* **117**, 564-575 (2023).
6. Yuan, S. *et al.* Health effects of high serum calcium levels: Updated phenome-wide Mendelian randomisation investigation and review of Mendelian randomisation studies. *EBioMedicine* **76**, 103865 (2022).